# Reciprocal action of Casein Kinase Iε on core planar polarity proteins regulates clustering and asymmetric localisation

Helen Strutt*, Jessica Gamage, David Strutt*

Department of Biomedical Science, University of Sheffield, Sheffield, United Kingdom

**Abstract** The conserved core planar polarity pathway is essential for coordinating polarised cell behaviours and the formation of polarised structures such as cilia and hairs. Core planar polarity proteins localise asymmetrically to opposite cell ends and form intercellular complexes that link the polarity of neighbouring cells. This asymmetric segregation is regulated by phosphorylation through poorly understood mechanisms. We show that loss of phosphorylation of the core protein Strabismus in the *Drosophila* pupal wing increases its stability and promotes its clustering at intercellular junctions, and that Prickle negatively regulates Strabismus phosphorylation. Additionally, loss of phosphorylation of Dishevelled – which normally localises to opposite cell edges to Strabismus – reduces its stability at junctions. Moreover, both phosphorylation events are independently mediated by Casein Kinase Iε. We conclude that Casein Kinase Iε phosphorylation acts as a switch, promoting Strabismus mobility and Dishevelled immobility, thus enhancing sorting of these proteins to opposite cell edges.
DOI: https://doi.org/10.7554/eLife.45107.001

*For correspondence:
h.strutt@sheffield.ac.uk (HS);
d.strutt@sheffield.ac.uk (DS)

Competing interests: The authors declare that no competing interests exist.

## Introduction

Phosphorylation is a widespread means of controlling protein activity, regulating protein-protein interactions, protein stability and conformation (*Hunter, 2007*). The activity of most signalling pathways is regulated by phosphorylation of pathway components. This includes the 'core' planar polarity pathway (*Singh and Mlodzik, 2012*; *Butler and Wallingford, 2017*): however, compared to other signalling pathways, the molecular mechanisms are poorly understood.

The core planar polarity proteins (hereafter, the 'core proteins') regulate the production of polarised structures or polarised cell behaviours in the plane of a tissue. This includes polarised production of cilia and of stereocilia bundles in the inner ear, and the coordinated polarisation of tissue movements necessary for convergence and extension of the body axis (*Devenport, 2016*; *Davey and Moens, 2017*; *Butler and Wallingford, 2017*). In *Drosophila*, the core pathway controls the production of polarised hairs and bristles on many adult tissues, for example the trichomes that emerge from the distal edge of each cell in the adult wing.

The core pathway specifies polarised structures via the asymmetric localisation of pathway components. In the *Drosophila* pupal wing, the seven-pass transmembrane protein Frizzled (Fz), and the cytoplasmic proteins Dishevelled (Dsh) and Diego (Dgo) localise to distal cell ends, where the trichome will emerge. The four-pass transmembrane protein Strabismus (Stbm, also known as Van Gogh [Vang]) and Prickle (Pk) localise to proximal cell ends, and the atypical cadherin Flamingo (Fmi, also known as Starry Night [Stan]) localises to both proximal and distal cell ends (*Figure 1A*). Fmi mediates homophilic adhesion that is important for coupling polarity between cells (reviewed in *Goodrich and Strutt, 2011*; *Devenport, 2016*; *Butler and Wallingford, 2017*).

**Figure 1.** Planar polarity and the cloud model of core protein localisation. (**A**) Core polarity proteins localise to proximal or distal edges of pupal wing cells (left), where they form intercellular complexes (right). (**B**) Live image of a 28 hr APF pupal wing expressing Stbm-EGFP. Asymmetrically localised core proteins cluster into membrane subdomains (puncta, yellow circle). The cyan circle indicates a non-puncta domain on the proximal-distal cell edges. (**C**) Diagram illustrating the possible organisation of the core polarity proteins. In non-puncta junctional regions, complexes associate at low density, in both orientations, and have relatively high mobility (left). Feedback interactions between the core proteins leads to complex sorting and complexes align in the same orientation. This promotes higher order multimerisation (red connectors) and reduced mobility (middle and right). (**D**) Diagram illustrating the position of the conserved phosphorylation site clusters (orange boxes) in the Stbm protein. The positions of the four transmembrane domains (black boxes, TM1-4) are also shown.

DOI: https://doi.org/10.7554/eLife.45107.002

The overall direction of polarisation is determined by tissue-specific global cues (*Aw and Devenport, 2017*). Polarity is then thought to be refined and amplified by feedback interactions between the core proteins. Mathematical modelling has suggested that feedback may involve destabilisation of complexes of opposite orientation and/or stabilisation of complexes in the same orientation. This can lead to sorting of complexes such that they all align in the same direction (*Amonlirdviman et al., 2005*; *Le Garrec et al., 2006*; *Burak and Shraiman, 2009*; *Schamberg et al., 2010*).

With regard to possible stabilising mechanisms, core protein asymmetry is associated with clustering of proteins into punctate membrane subdomains (*Figure 1B*, *Strutt et al., 2011*; *Cho et al., 2015*) and reduced core protein turnover (*Strutt et al., 2011*; *Butler and Wallingford, 2015*; *Chien et al., 2015*; *Strutt et al., 2016*). Based on a detailed study of core protein organisation in puncta, we recently proposed that core proteins form a non-stoichiometric 'cloud' around a Fmi-Fz nucleus (*Strutt et al., 2016*). Feedback interactions lead to sorting of complexes, and multiple protein-protein interactions are thought to promote a phase transition into higher order 'signalosome-like' structures, where arrays of complexes of the same orientation are stabilised (*Figure 1C*,

*Strutt et al., 2016*). Interestingly, Stbm stoichiometry was found to be much higher than that of the other core proteins (*Strutt et al., 2016*). The reasons for this are unclear, but could relate to a role for Stbm in promoting higher order structures. Furthermore, Pk may stabilise Stbm by promoting complex clustering (*Tree et al., 2002*; *Bastock et al., 2003*; *Cho et al., 2015*).

Mechanisms of destabilisation may include competitive binding between core proteins (*Tree et al., 2002*; *Carreira-Barbosa et al., 2003*; *Jenny et al., 2005*; *Amonlirdviman et al., 2005*). More specifically, Pk (a 'proximal' complex component) is known to destabilise Fz and/or Dsh ('distal' components) in the same cell (*Warrington et al., 2017*). In addition, Pk has been suggested to destabilise complexes containing Stbm and Fmi (*Cho et al., 2015*). However, knowledge of additional molecular mechanisms by which core proteins might become destabilised or clustered together is very poor, and post-translational modifications such as phosphorylation are likely to be a key element.

Indeed, core protein phosphorylation is essential for feedback amplification of asymmetry. In particular, reduced activity of Casein Kinase Iε (CKIε, also known as Discs Overgrown [Dco] or Doubletime [Dbt] in flies) causes planar polarity defects and a reduction in core protein asymmetry (*Strutt et al., 2006*; *Klein et al., 2006*; *Kelly et al., 2016*; *Yang et al., 2017*). Interestingly, CKIε has been implicated in phosphorylation of both Stbm and Dsh. CKIε was first found to bind to and phosphorylate the vertebrate Dsh homologue (Dvl) in canonical Wnt signalling (*Peters et al., 1999*; *McKay et al., 2001*). In planar polarity in flies, Dsh phosphorylation correlates with its recruitment to cellular junctions by Fz (*Axelrod, 2001*; *Shimada et al., 2001*), where it is incorporated into stable complexes (*Strutt et al., 2016*), and decreased Dsh phosphorylation is seen in *dco* mutants (*Strutt et al., 2006*).

The exact phosphorylation sites for CKIε in Dsh/Dvl are not well defined, but a mutation of a serine/threonine-rich region upstream of the PDZ domain affects Dvl recruitment to membranes in *Xenopus* (*Ossipova et al., 2005*). Moreover, mutation of one of these residues (S236 in fly Dsh) blocks phosphorylation of Dsh by Dco in vitro (*Klein et al., 2006*). However, a transgene in which these residues were mutated largely rescued the planar polarity defects of *dsh* mutants in the adult fly wing (*Strutt et al., 2006*; but see also *Penton et al., 2002*).

More recently, CKIε has been implicated in phosphorylating Stbm and its vertebrate homologue Vangl2 (*Gao et al., 2011*; *Kelly et al., 2016*; *Yang et al., 2017*). In particular, Wnt gradients were proposed to lead to a gradient of Vangl2 phosphorylation and asymmetry in the vertebrate limb (*Gao et al., 2011*). CKIε promotes Stbm/Vangl2 phosphorylation in cell culture (*Gao et al., 2011*; *Kelly et al., 2016*; *Yang et al., 2017*). Two clusters of conserved serine and threonine residues were identified as CKIε phosphorylation sites. Mutation of some or all of these residues leads to a loss of Stbm/Vangl2 phosphorylation in cell culture, and defects in planar polarisation (*Gao et al., 2011*; *Ossipova et al., 2015*; *Kelly et al., 2016*; *Yang et al., 2017*).

The fact that CKIε has been implicated in phosphorylating both Stbm/Vangl2 and Dsh/Dvl in cell culture leads to the question of whether both proteins are bona fide targets in vivo. For instance, both Fz and Dsh/Dvl have been proposed to promote Stbm/Vangl2 phosphorylation by CKIε (*Kelly et al., 2016*; *Yang et al., 2017*). Thus, it is possible that only Stbm/Vangl2 are direct targets of CKIε and that Stbm/Vangl2 phosphorylation has a secondary effect on Fz-Dsh/Dvl behaviour. Moreover, mechanistic insight into how these phosphorylation events affect core protein sorting and asymmetry is lacking.

Here, we demonstrate that CKIε has independent and reciprocal actions on Dsh and Stbm during planar polarity signalling in *Drosophila*. We use phosphorylation site mutations in Stbm to show that lack of Stbm phosphorylation leads to its clustering in 'mixed' puncta that contain complexes in both orientations. CKIε-dependent phosphorylation increases Stbm turnover at junctions, and thus promotes complex sorting, while phosphorylation of Dsh decreases its turnover. Pk negatively regulates Stbm phosphorylation and increases Stbm stability. These results support a direct role for Dco in phosphorylating both Stbm and Dsh in vivo in planar polarity signalling.

## Results

### Stbm phosphorylation sites are essential for core protein asymmetry

Previous work identified two conserved clusters of serine and threonine residues within vertebrate Vangl2, which are phosphorylated in tissue culture (*Figure 1D*, *Gao et al., 2011*; *Yang et al., 2017*). *P[acman]-stbm* rescue constructs (*Strutt et al., 2016*) were generated, in which all serine/threonine residues in clusters I and II were mutated to alanine (phosphomutant 'S[All]A') or glutamic acid (phosphomimetic 'S[All]E'). These residues are in regions of Stbm predicted to be unstructured (data not shown), so this was not expected to alter the secondary structure of Stbm. Neither the phospho-mutant nor the phosphomimetic form of Stbm rescued the trichome orientation of *stbm* null mutants, while wild-type *P[acman]-stbm* in the same genomic site gave complete rescue (*Figure 2A–D* and *Figure 2—figure supplement 1D*). The failure of the phosphomimetic version to rescue may be because glutamic acid does not completely substitute for phosphorylated serine and threonine residues within Stbm. Alternatively, Stbm may need to cycle between phosphorylated and unphosphorylated forms in order to function in planar polarity, and the mutated proteins are unable to perform this cycling.

Core protein localisation in phosphomutant and phosphomimetic pupal wings was then examined. Twin clones were made, in which tissue expressing wild-type Stbm was juxtaposed to tissue expressing mutant forms of Stbm, both in the absence of endogenous *stbm* gene activity. In keeping with the strong trichome orientation defects, a strong decrease in core protein asymmetry was seen in pupal wings expressing either phosphomutant or phosphomimetic forms of Stbm (*Figure 2E–H*, *Figure 2—figure supplement 1H and I*, *Figure 2—source data 1*). There was also a slight increase in overall levels of phosphomutant or phosphomimetic Stbm at cellular junctions, compared to wild-type Stbm (*Figure 2I*, *Figure 2—source data 1*). We conclude that the phosphorylation sites in Stbm are necessary for its correct asymmetric localisation and to orient trichomes in the adult wing.

Western blotting of pupal wing extracts confirmed that endogenous Stbm is phosphorylated in vivo, with the majority of protein existing in a phosphorylated state (*Figure 2—figure supplement 2A*). As expected, the phosphomutant form showed increased mobility on SDS-PAGE, while the phosphomimetic form had a similar mobility to that of endogenous Stbm (*Figure 2J*). However, as expected, the mobility of phosphomimetic Stbm was not sensitive to phosphatase treatment (*Figure 2—figure supplement 2A*). Overall cellular protein levels were similar to wild-type (*Figure 2K*, *Figure 2—source data 1*).

It has been suggested that residue 5 in cluster II, and residues 120 and 122 in cluster I are 'founder sites', such that phosphorylation on these leads to a cascade of phosphorylation on neighbouring residues (*Gao et al., 2011*; *Ossipova et al., 2015*; *Kelly et al., 2016*; *Yang et al., 2017*). However, *P[acman]-stbm* constructs simultaneously mutant for all three founder sites fully rescued trichome polarity in adult wings, and core protein asymmetry in pupal wings was normal (*Figure 2—figure supplement 1*, *Figure 2—source data 1*). This contrasts with the work of *Kelly et al. (2016)*, who reported planar polarity defects in flies after mutation of serines 120 and 122 to alanine. The difference in our results could be due to abnormal or uneven expression of the *tub-StbmS2A* rescue construct used in *Kelly et al. (2016)*. Furthermore, mutation of all phosphorylation sites in only cluster I or cluster II revealed that the phosphorylation sites within cluster I are sufficient for correct core protein asymmetry, and are responsible for most of the retardation in mobility on SDS-PAGE (*Figure 2—figure supplement 2B–H*, *Figure 2—source data 1*).

### Uncoupling of puncta formation and asymmetry in Stbm phosphomutants

Interestingly, previous data has suggested a correlation between core protein asymmetric localisation to opposite cell ends, and the formation of large junctional puncta (*Strutt et al., 2011*; *Cho et al., 2015*). In keeping with this idea, fewer puncta were observed in Stbm phosphomimetic flies, and non-puncta material also increased (*Figure 2F*). However, in the Stbm phosphomutant, Stbm still appeared to cluster into puncta, despite the loss of asymmetry (*Figure 2E*).

To quantitate puncta size, we thresholded images using the same threshold value in wild-type and mutant regions of the same wings. Fmi co-immunolabelling was used to select puncta, as overall junctional levels of Fmi do not change in either the phosphomutant or the phosphomimetic tissue

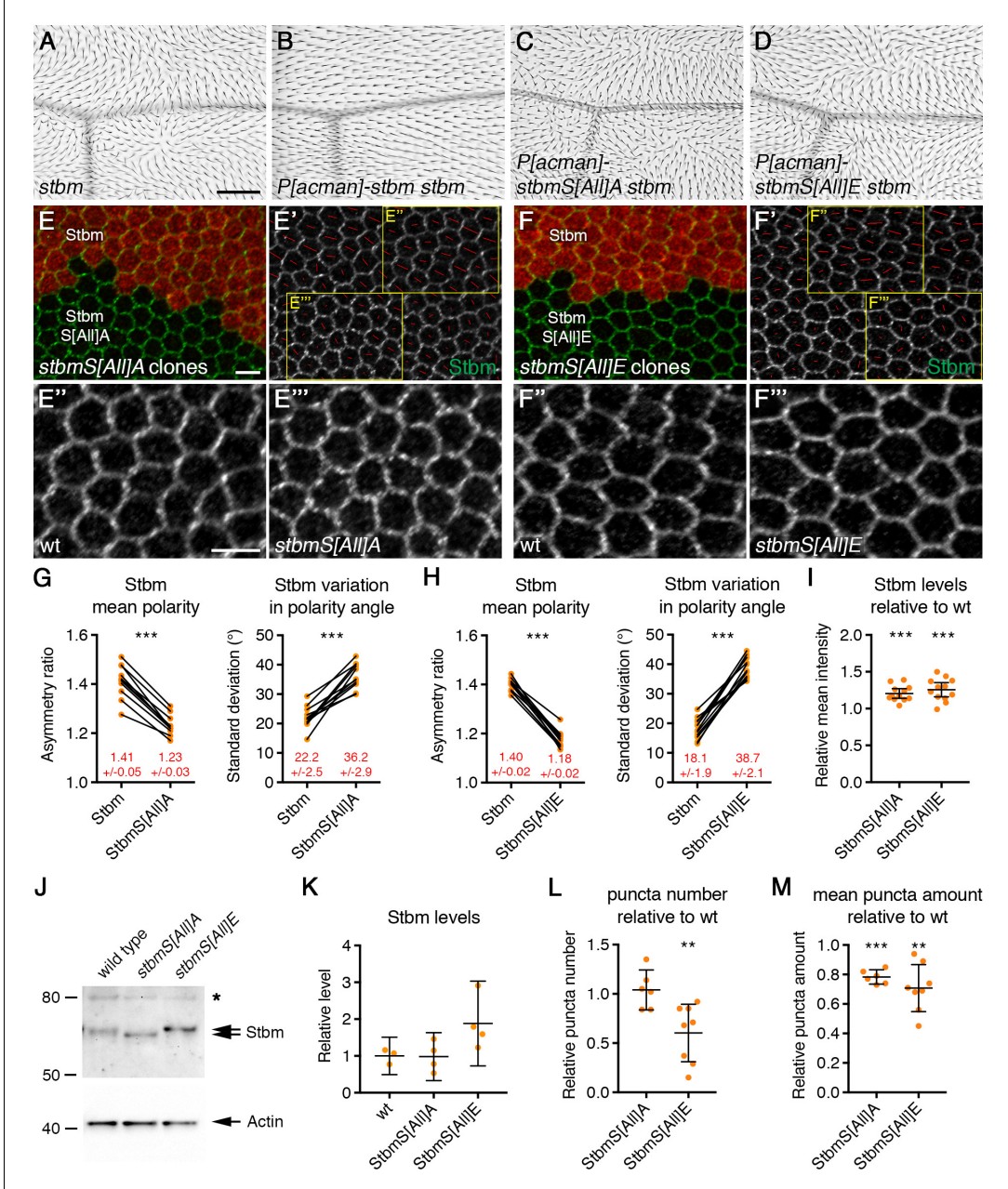

**Figure 2.** Disrupted trichome orientation in Stbm phosphomutants and phosphomimetics. (A–D) Adult wings from *stbm^6* mutant flies (A), *P[acman]-stbm stbm^6* flies (B), *P[acman]-stbmS[All]A stbm^6* flies (C), or *P[acman]-stbmS[All]E stbm^6* flies (D). Scale bar 100 µm. (E,F) 28 hr APF pupal wings, carrying twin clones of *P[acman]-stbm stbm^6*, marked by β-gal immunolabelling (red), next to *P[acman]-stbmS[All]A stbm^6* (E) or *P[acman]-stbmS[All]E stbm^6* (F). Wings immunolabelled for Stbm in green. (E',F') Stbm immunolabelling overlaid with polarity nematics (red lines), where the length of line indicates mean cell polarity and the orientation indicates direction of polarity. Yellow boxes indicate zoomed regions shown in E'' and F'' (wild-type regions) or E''' and F''' (mutant regions). Scale bar 5 µm. (G,H) Quantitation of mean polarity and variation in polarity angle, of 28 hr APF pupal wings immunolabelled for Stbm in twin clones of *P[acman]-stbm stbm^6* and *P[acman]-stbmS[All]A stbm^6* (G, n = 11) or *P[acman]-stbmS[All]E stbm^6* (H, n = 13). Values from the same wing are linked by black bars, mean and 95% confidence intervals are listed. Paired t-tests were used to compare values in the same wings, ***p≤0.001. (I) Quantitation of mean intensity of Stbm immunolabelling at junctions of 28 hr APF pupal wings, shown as a ratio of signal in *P[acman]-stbmS[All]A stbm^6* (n = 12) or *P[acman]-stbmS[All]E stbm^6* (n = 12) compared to *P[acman]-stbm stbm^6* (wt) in each wing. Error bars are 95% confidence intervals. One-sample t-tests were used to determine if the ratio differed from 1.0, ***p≤0.001. (J) Western blot probed with Stbm antibody, of extracts from 28 hr APF pupal wings from wild-type, *P[acman]-stbmS[All]A stbm^6* or *P[acman]-stbmS[All]E stbm^6* flies. The asterisk indicates a non-specific band. Actin was used as a loading control. (K) Quantitation of Stbm levels from western blots, from wild-type (n = 3), *P[acman]-stbmS[All]A stbm^6* (n = 4) or *P[acman]-stbmS[All]E stbm^6* (n = 4) pupal wings. Error bars are 95% confidence intervals. Levels were compared to wild-type by ANOVA with Dunnett's multiple comparisons test, no significant differences were seen. (L,M) Quantitation of puncta number (L) and mean puncta

*Figure 2 continued on next page*

*Figure 2 continued*

amount (M), from 28 hr APF pupal wings immunolabelled for Stbm, shown as a ratio of signal in *P[acman]-stbmS[All]A stbm⁶* (n = 6) or *P[acman]-stbmS [All]E stbm⁶* (n = 8) compared to *P[acman]-stbm stbm⁶* (wt) in each wing. Puncta were detected using Fmi immunolabelling. Error bars are 95% confidence intervals. One-sample t-tests were used to determine if the ratio differed from 1.0, **≤0.01, ***p≤0.001.
DOI: https://doi.org/10.7554/eLife.45107.003

The following source data and figure supplements are available for figure 2:

**Source data 1.** Quantification of Stbm levels, asymmetry and puncta size in Stbm phosphomutant and phosphomimetic wings.
DOI: https://doi.org/10.7554/eLife.45107.006
**Figure supplement 1.** Putative founder site mutants do not disrupt core protein asymmetry.
DOI: https://doi.org/10.7554/eLife.45107.004
**Figure supplement 2.** Dissection of Stbm phosphorylation site clusters.
DOI: https://doi.org/10.7554/eLife.45107.005

(see below). This revealed that a similar number of puncta were seen in the phosphomutant as in wild-type tissue, while many fewer puncta were seen in the phosphomimetic (*Figure 2L*, *Figure 2— source data 1*). The mean amount of Stbm in puncta was however slightly reduced in both phosphomutant and phosphomimetic tissue (*Figure 2M*, *Figure 2—source data 1*). As the phosphomutant forms a similar number of puncta to wild-type, this suggests that the coupling between puncta formation and core protein asymmetry is lost in Stbm phosphomutants.

Other core proteins co-localised with both phosphomutant and phosphomimetic Stbm (*Figure 3A–D*). Overall levels of Fmi and Dsh in junctions were similar to wild-type, while Fz levels were slightly decreased in both cases, and Pk levels were lower in phosphomutant and higher in phosphomimetic wings (*Figure 3—figure supplement 1A–J*, *Figure 3—source data 1*).

## Stbm phosphomutant puncta contain complexes in both orientations

Core protein complexes within junctional puncta are highly polarised compared to other junctional regions (*Strutt et al., 2011*; *Cho et al., 2015*; *Strutt et al., 2016*). This is consistent with puncta containing arrays of core protein complexes, all aligned in the same direction (*Figure 1C*, right). As Stbm phosphomutant puncta are no longer associated with overall asymmetry we asked whether individual puncta are still polarised, or whether phosphomutant puncta have a different organisation. To test this, we made adjacent twin clones of EGFP-tagged Dgo next to mApple-tagged Dgo. In both wild-type and phosphomutant backgrounds, EGFP-Dgo and mApple-Dgo co-localise with other core proteins in puncta, as expected (*Figure 3—figure supplement 1K and L*). In a wild-type background, Dgo localised predominantly to distal cell ends, as previously reported (*Das et al., 2004*), so puncta on clone boundaries contained either EGFP-Dgo (green) or mApple-Dgo (red) (*Figure 3E,G*). We could envisage two scenarios for Stbm phosphomutant puncta on clone boundaries: if individual puncta were polarised, we would expect to see puncta containing either EGFP-Dgo or mApple-Dgo, but Dgo in puncta could localise to any cell edge (*Figure 3F*, left). Alternatively if individual puncta were not polarised, co-localisation of EGFP-Dgo and mApple-Dgo would be seen (*Figure 3F*, right). Such co-localisation of EGFP-Dgo and mApple-Dgo was indeed observed (*Figure 3H*). This indicates that individual puncta (at least at this optical resolution) contain complexes in both orientations (*Figure 3I*), and that inhibition of Stbm phosphorylation disrupts sorting of complexes.

## Phosphorylation of Stbm regulates its turnover at junctions

In a wild-type situation, the alignment of core proteins in the same orientation within puncta correlates with low protein turnover, as measured by Fluorescence Recovery After Photobleaching (FRAP) assays (*Strutt et al., 2011*). As the Stbm phosphomutant forms abnormal puncta with complexes in both orientations, we investigated the turnover of phosphomutant Stbm. A 'hub-and-spoke' FRAP methodology was used, in which the junctions in the equivalent of half a cell are bleached (*Warrington et al., 2017*, *Figure 4A*). This avoids excessive bleaching of total protein in any single cell, while allowing junctions of all orientations to be sampled, regardless of whether they are enriched for core proteins or contain puncta.

Hub-and-spoke FRAP showed that the stable amount of phosphomutant Stbm-EGFP at junctions was increased compared to wild-type Stbm-EGFP, while the unstable amount was unchanged

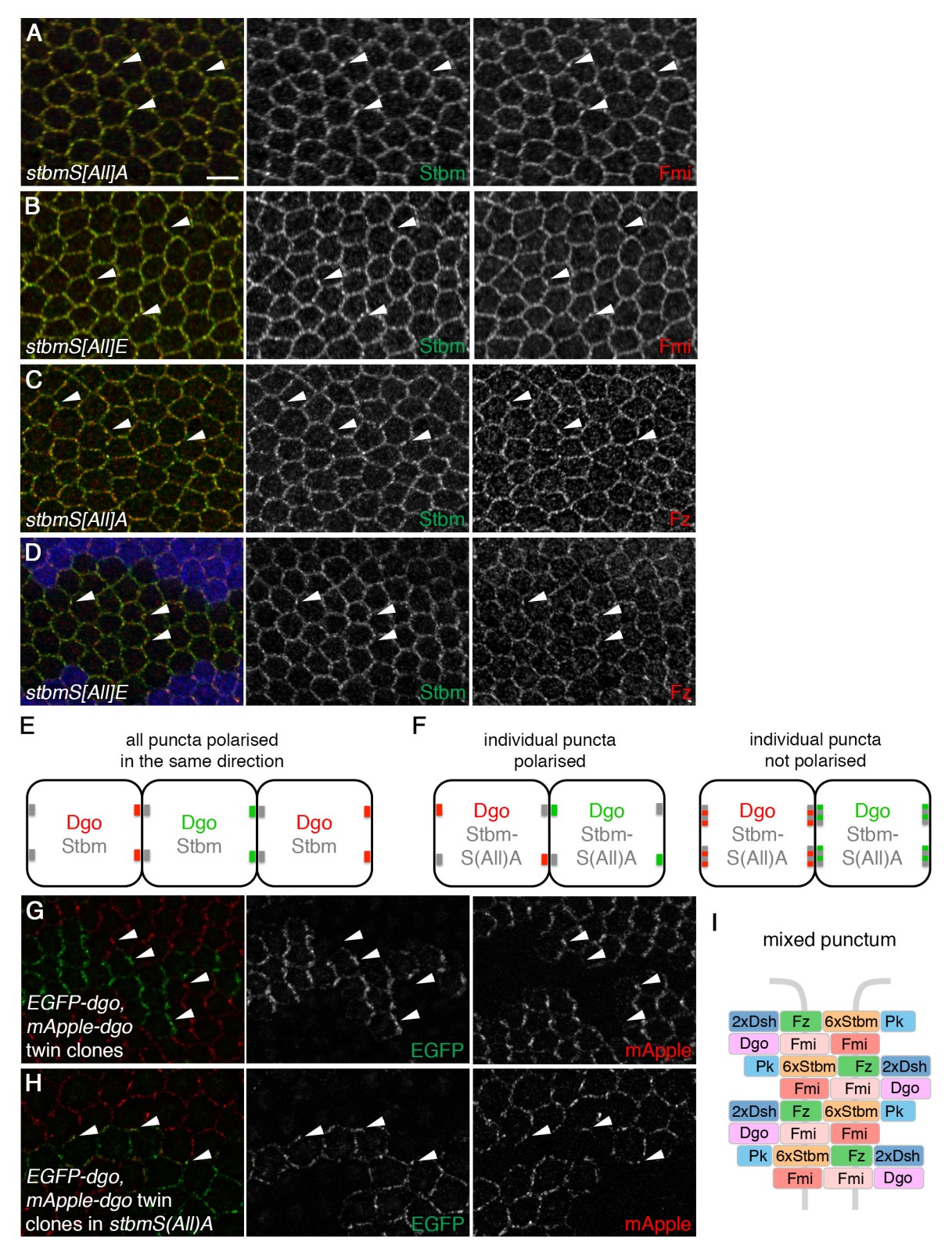

**Figure 3.** Stbm phosphomutants form 'mixed' puncta. (A–D) 28 hr APF pupal wings expressing *P[acman]-stbmS[All]A stbm^6* (A,C) or *P[acman]-stbmS [All]E stbm^6* (B,D). Blue immunolabelled tissue in (D) is wild-type. Wings immunolabelled for Stbm in green and Fmi (A,B) or Fz (C,D) in red. Arrowheads point to examples of puncta in which Stbm and Fmi or Fz co-localise. Scale bar 5 µm. (E,F) Schematic of twin clone experiment. Groups of cells express either EGFP-Dgo (green) or mApple-Dgo (red), and puncta are examined at the boundaries between them. (E) In wild-type wings, Dgo localises to

*Figure 3 continued on next page*

*Figure 3 continued*
distal cell ends, so puncta on distal clone boundaries contain either only EGFP-Dgo or only mApple-Dgo. (**F**) In phosphomutant wings, Dgo is not asymmetrically localised. If individual puncta are polarised (left), puncta on clones boundaries will contain either EGFP-Dgo or mApple-Dgo, regardless of whether the boundary is proximal or distal. If individual puncta are not polarised (right), puncta will contain both EGFP-Dgo and mApple-Dgo. (**G,H**) 28 hr APF pupal wings carrying twin clones of *P[acman]-EGFP-dgo dgo^{380}* next to *P[acman]-mApple-dgo dgo^{380}*, in a wild-type (**G**) or a *P[acman]-stbmS [All]A stbm^{6}* mutant background (**H**). EGFP fluorescence is in green and mApple fluorescence is in red. White arrowheads indicate specific puncta on clone boundaries. Puncta are labelled with either green or red Dgo in wild-type, but in *stbmS[All]A* tissue, puncta on clone boundaries appear yellow, as they contain both green and red Dgo. (**I**) Schematic of 'mixed' punctum, containing core protein complexes in both orientations (compare to *Figure 1C*, right).
DOI: https://doi.org/10.7554/eLife.45107.007
The following source data and figure supplement are available for figure 3:

**Source data 1.** Quantification of core protein levels in Stbm phosphomutant and phosphomimetic wings.
DOI: https://doi.org/10.7554/eLife.45107.009
**Figure supplement 1.** Core protein levels and localisation in Stbm phosphomutant and phosphomimetic wings.
DOI: https://doi.org/10.7554/eLife.45107.008

(*Figure 4B*, *Figure 4—figure supplement 1*, *Figure 4—source data 1*). This suggests that Stbm phosphorylation promotes turnover of Stbm. Consistent with this, the stable amount of phosphomimetic Stbm-EGFP did not change compared to wild-type, but the unstable amount increased (*Figure 4B*, *Figure 4—figure supplement 1*, *Figure 4—source data 1*). This suggests that phosphomimetic Stbm accumulates at junctions even when not stably incorporated into complexes.

Taken together with our previous data that there is less phosphomimetic Stbm incorporated into puncta (*Figure 2L,M*), this supports a model in which phosphorylation promotes Stbm turnover, while lack of phosphorylation promotes Stbm clustering in complexes (*Figure 4C*). As both protein turnover and clustering are thought to be required for sorting of core proteins to opposite cell ends, these data are consistent with the idea that Stbm normally cycles between phosphorylated and unphosphorylated states during complex sorting and establishment of asymmetry.

We also measured the turnover of Stbm *within* puncta (*Figure 4D*), as these are normally sites of high core protein asymmetry and stability, but contain complexes of mixed orientation in Stbm phosphomutants. In keeping with the hub-and-spoke FRAP, an increase in the stable fraction of phosphomutant Stbm-EGFP (i.e. decreased turnover) was seen in puncta, while the stable fraction was decreased in phosphomimetic Stbm-EGFP puncta (*Figure 4E*, *Figure 4—source data 1*, note that as puncta size varies between genotypes we were not able to translate stable fractions into stable amounts).

As core protein complexes are thought to be sorted via feedback interactions, we considered how the altered stability of Stbm phosphomutants and phosphomimetics affected the localisation and stability of other core proteins. Interestingly, the stable fraction of Fz-EGFP and Fmi-EGFP within puncta was decreased in both a Stbm phosphomutant and a Stbm phosphomimetic background (*Figure 4F and G*, *Figure 4—source data 1*). We interpret this to mean that inhibiting Stbm phosphorylation promotes excess clustering and stability of Stbm within complexes; but the presence of oppositely oriented complexes may promote negative feedback interactions, leading to destabilisation of other complex components. In phosphomutant Stbm wings, this competition between complex stabilisation and destabilisation results in a net increase in Stbm stability, but a net decrease in Fmi and Fz stability. In the phosphomimetic, the result is a net decrease in stability of all three core proteins. In both cases negative feedback between unsorted complexes may prevent puncta growing to the same size as wild-type puncta (see *Figure 2M*).

A role for Stbm in promoting clustering of complexes of the same orientation may be a mechanism for feedback amplification of asymmetry. Interestingly, wings expressing one copy of either phosphomutant or phosphomimetic Stbm and one copy of wild-type Stbm failed to polarise, suggesting that both mutant forms act as dominant negatives (*Figure 4—figure supplement 2*, *Figure 4—source data 1*, see also *Yang et al., 2017*). Large puncta were observed in the phosphomutant heterozygotes (*Figure 4—figure supplement 2B*), consistent with a model in which excess clustering of phosphomutant Stbm leads to recruitment of wild-type Stbm into abnormal, mixed puncta.

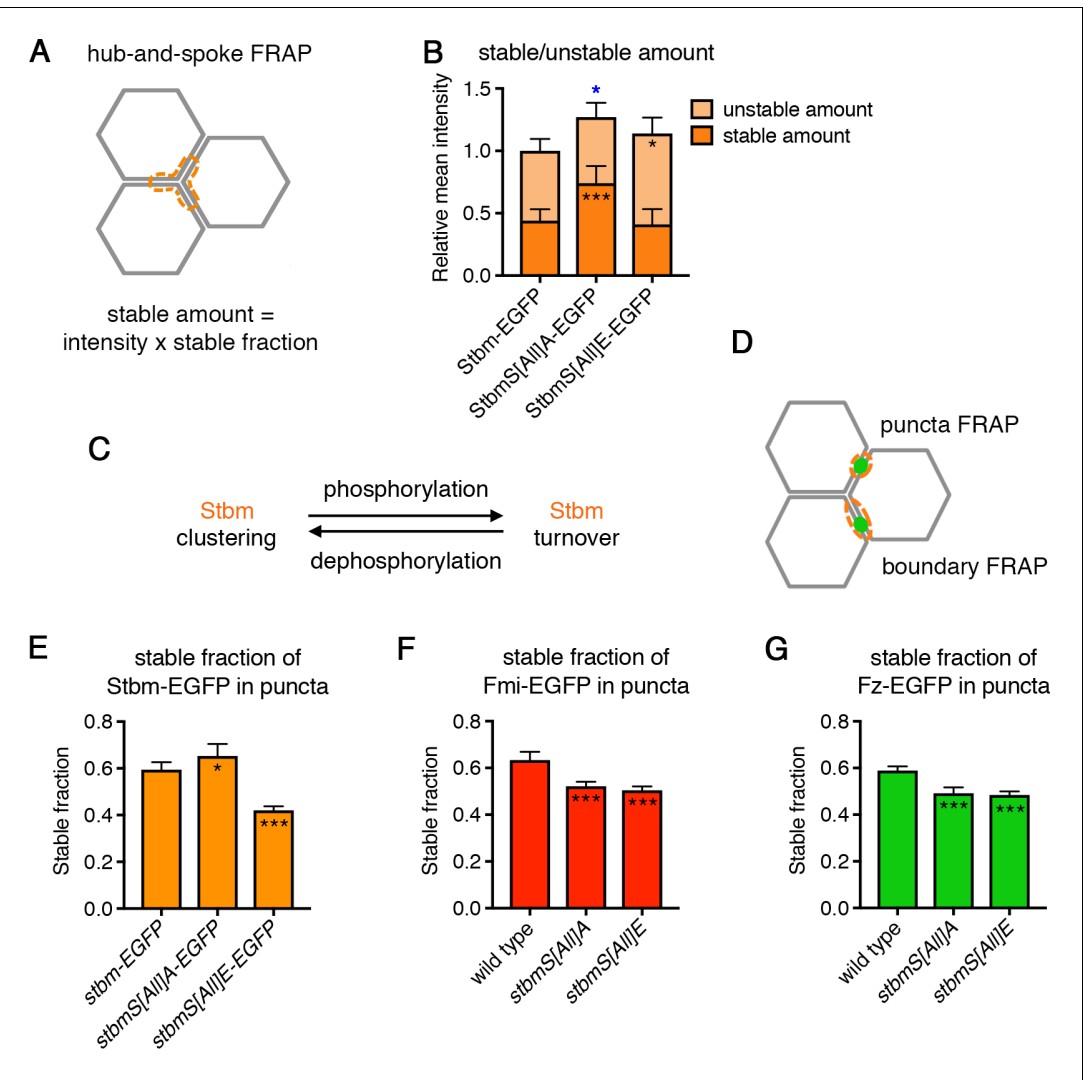

**Figure 4.** Stbm phosphomutants have increased stability at junctions. (**A**) Schematic of hub-and-spoke FRAP. The orange dotted line shows the 'hub-and-spoke' region that is bleached, which corresponds to three cell vertices and the equivalent of three cell junctions. The stable fraction, as determined by FRAP, is multiplied by the total initial intensity within the bleached region to give a stable amount of protein at junctions for each wing. (**B**) Stable and unstable amounts of EGFP-tagged protein in 28 hr APF pupal wings after hub-and-spoke FRAP. Flies were *P [acman]-stbm-EGFP stbm⁶/+* (n = 11), *P[acman]-stbmS[All]A-EGFP stbm⁶/P[acman]-stbmS[All]A stbm⁶* (n = 9) and *P [acman]-stbmS[All]E-EGFP stbm⁶/P[acman]-stbmS[All]E stbm⁶* (n = 11). Data are normalised to the total intensity in Stbm-EGFP. Error bars are 95% confidence intervals, and ANOVA with Dunnett's multiple comparisons test was used to compare stable amounts (asterisks in dark orange columns), unstable amounts (asterisks in light orange columns) or total amounts (blue asterisks above the columns) to the Stbm-EGFP control, p≤0.05*, p≤0.001***. (**C**) Summary diagram showing the effect of phosphorylation and dephosphorylation on Stbm turnover. (**D**) Schematic of puncta FRAP and boundary FRAP. In puncta FRAP, an elliptical region surrounding a punctum is bleached (***Figure 4E–G***), whilst in boundary FRAP an entire junction on a clone boundary is bleached (***Figure 6G***). Note that as puncta of different genotypes are different sizes, stable fractions in puncta FRAP cannot be translated into stable amounts. (**E–G**) Stable fraction of EGFP-tagged protein in puncta in 28 hr APF pupal wings. (**E**) *P[acman]-stbm-EGFP stbm⁶* (n = 6), *P[acman]-stbmS[All]A-EGFP stbm⁶* (n = 7), *P[acman]-stbmS[All]E-EGFP stbm⁶* (n = 9). (**F**) *fmi-EGFP/+* (n = 9), *P[acman]-stbmS[All]A stbm⁶/P[acman]/stbmS[All]A stbm⁶ fmi-EGFP* (n = 10), *P[acman]-stbmS [All]E stbm⁶/P[acman]/stbmS[All]E stbm⁶ fmi-EGFP* (n = 10). (**G**) *fz-EGFP/+* (n = 9), *P[acman]-stbmS[All]A stbm⁶; fz-EGFP/+* (n = 8), *P[acman]-stbmS[All]E stbm⁶; fz-EGFP/+* (n = 7). The fluorescence recovery was fitted to an exponential curve for each genotype, and the graph shows the stable fraction (1-$Y^{max}$) and the 95% confidence intervals. Stable fractions were compared to control (wild-type Stbm or Stbm-EGFP) using an extra sum of squares F test, p≤0.05*, p≤0.001***.

*Figure 4 continued on next page*

*Figure 4 continued*

DOI: https://doi.org/10.7554/eLife.45107.010

The following source data and figure supplements are available for figure 4:

**Source data 1.** Quantification of Stbm phosphomutant and phosphomimetic stability, and dominant negative effects.

DOI: https://doi.org/10.7554/eLife.45107.013

**Figure supplement 1.** FRAP analysis of Stbm phosphomutant and phosphomimetic wings.

DOI: https://doi.org/10.7554/eLife.45107.011

**Figure supplement 2.** Dominant negative effects of Stbm phosphomutant and phosphomimetic.

DOI: https://doi.org/10.7554/eLife.45107.012

## In vivo regulation of Stbm by Dco kinase

Experiments in cultured cells have suggested that at least some of the phosphorylation of Stbm or Vangl2 could be mediated by the kinase CKIε (Dco in flies, *Gao et al., 2011*; *Kelly et al., 2016*; *Yang et al., 2017*). However, this has proven difficult to verify in vivo. Stbm migration on SDS-PAGE was not altered in *dco* hypomorphs (*Figure 5—figure supplement 1A*, see also *Kelly et al., 2016*). However, we did see a subtle increase in Stbm migration after expression of dominant-negative Dco in pupal wings (*Figure 5—figure supplement 1B*).

To confirm a role for Dco in regulating Stbm phosphorylation, we examined the turnover of Stbm by FRAP when either dominant-negative or wild-type Dco were overexpressed. Overexpression of dominant-negative Dco led to an increase in the stable amount of Stbm-EGFP at junctions,

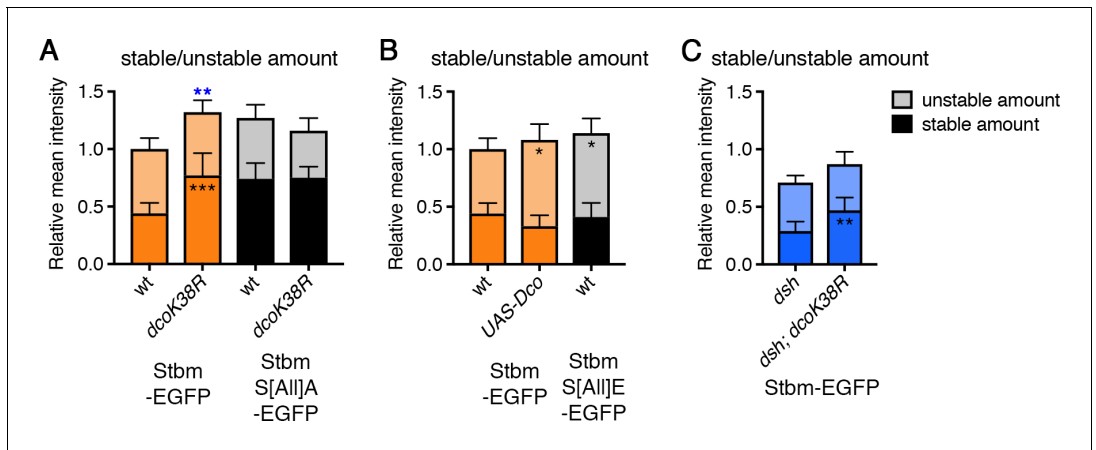

**Figure 5.** Regulation of Stbm phosphorylation and turnover by Dco. (**A–C**) Stable and unstable amounts of EGFP-tagged protein, in 28 hr APF pupal wings after hub-and-spoke FRAP. Flies were (**A**) *P[acman]-stbm-EGFP stbm[6]/+* (n = 11), *P[acman]-stbm-EGFP stbm[6]/en-GAL4; UAS-dco[K38R]/+* (n = 11), *P[acman]-stbmS[All]A-EGFP stbm[6]/P[acman]-stbmS[All]A stbm[6]* (n = 9) and *P[acman]-stbmS[All]A-EGFP stbm[6]/P[acman]-stbmS[All]A stbm[6] en-GAL4; UAS-dco[K38R]/+* (n = 15); (**B**) *P[acman]-stbm-EGFP stbm[6]/+* (n = 11), *UAS-Dco/+; P[acman]-stbm-EGFP-stbm[6]/en-GAL4* (n = 11) and *P[acman]-stbmS[All]E-EGFP stbm[6]/P[acman]-stbmS[All]E stbm[6]* (n = 11); (**C**) *dsh[1]; P[acman]-stbm-EGFP stbm[6]/+* (n = 12) and *dsh[1]; P[acman]-stbm-EGFP stbm[6]/en-GAL4; UAS-dco[K38R]/+* (n = 10). Wings were imaged in the posterior compartment. Data are normalised to the total intensity in Stbm-EGFP and error bars are 95% confidence intervals. Stable amounts (asterisks in dark shaded columns), unstable amounts (asterisks in light shaded columns) or total amounts (blue asterisks above the columns) were compared between genotypes, using ANOVA with Holm-Sidak's multiple comparisons test to compare pairs of samples with and without Dco[K38R] (**A**), ANOVA with Dunnett's multiple comparisons test to compare with the Stbm-EGFP control (**B**) or an unpaired t-test (**C**). p≤0.05*, p≤0.01**, p≤0.001***.

DOI: https://doi.org/10.7554/eLife.45107.014

The following source data and figure supplements are available for figure 5:

**Source data 1.** Quantification of Stbm stability in *dco* mutant wings.

DOI: https://doi.org/10.7554/eLife.45107.017

**Figure supplement 1.** Analysis of Stbm phosphorylation in *dco* mutant backgrounds.

DOI: https://doi.org/10.7554/eLife.45107.015

**Figure supplement 2.** FRAP analysis of Stbm in *dco* mutant backgrounds.

DOI: https://doi.org/10.7554/eLife.45107.016

phenocopying the results from Stbm phosphomutants (*Figure 5A*, *Figure 5—figure supplement 2A–C*, *Figure 5—source data 1*). Expression of dominant-negative Dco had no further effect on the stable amount of phosphomutant Stbm (*Figure 5A*, *Figure 5—figure supplement 2A–C*, *Figure 5—source data 1*). Conversely, overexpression of wild-type Dco, which would be expected to promote excess phosphorylation, caused an increase in the unstable amount of Stbm, similar to the Stbm phosphomimetic (*Figure 5B*, *Figure 5—figure supplement 2D–F*, *Figure 5—source data 1*).

As Dco is known to act on Dsh as well as Stbm, we then asked whether the effect of Dco on Stbm turnover was independent of any effect on Dsh. Overall levels of Stbm are decreased in a *dsh*[1] mutant background (a planar polarity-specific mutation), leading to a decrease in the absolute amount of stable Stbm (*Figure 6B*, *Figure 6—figure supplement 1C–E*, *Figure 6—source data 1*). However, expression of dominant-negative Dco in a *dsh* mutant still caused an increase in the stable amount of Stbm (*Figure 5C*, *Figure 5—figure supplement 2G–I*, *Figure 5—source data 1*). Therefore, we conclude that Dco affects Stbm turnover independently of Dsh and supports a model in which Dco regulates Stbm turnover by direct phosphorylation of Stbm.

## Stbm phosphorylation and turnover are negatively regulated by Pk

If Stbm phosphorylation normally controls Stbm turnover and this is important for core protein asymmetry, we considered the possibility that Stbm phosphorylation might be regulated by other core proteins. Interestingly, we saw a decrease in Stbm migration on SDS-PAGE in a *pk* mutant background (*Figure 6A*). This decrease in migration was lost after phosphatase treatment (*Figure 6—figure supplement 2A*), suggesting that Pk negatively regulates Stbm phosphorylation. Consistent with this, loss of Pk did not alter migration of either the Stbm phosphomutant or the phosphomimetic on SDS-PAGE (*Figure 6—figure supplement 2B*).

Overexpression of Pk causes excess clustering of core proteins into large junctional puncta (*Tree et al., 2002*; *Bastock et al., 2003*), which contain complexes in both orientations (*Figure 6—figure supplement 1A B*), as seen for Stbm phosphomutant puncta. Interestingly, this also led to increased migration of Stbm on SDS-PAGE, suggesting decreased phosphorylation (*Figure 6C*). Thus, Pk overexpression mimics Stbm phosphomutant phenotypes, while loss of Pk has the opposite effect.

Previous studies have suggested that Fz or Dsh might promote Stbm phosphorylation (*Kelly et al., 2016*; *Yang et al., 2017*). This would support a model in which Fz or Dsh recruit a kinase, thus phosphorylating and destabilising Stbm in complexes in the opposite orientation. However, we did not see any change in the migration of endogenous Stbm on SDS-PAGE in *fz* or *dsh* mutants, with Stbm still migrating more slowly than phosphomutant Stbm (*Figure 6A*). This differs from the work of *Kelly et al. (2016)*, who observed increased mobility of FLAG-tagged Stbm in *fz* mutants. We do not know why our results differ, but it is possible that tagging Stbm at the C-terminus with FLAG affects its function. Further support for our data comes from the observation that the hyperphosphorylation seen in *pk* mutants is retained in *pk; fz* or *dsh; pk* mutants, suggesting that neither Fz nor Dsh is needed for this hyperphosphorylation (*Figure 6E*). However, loss of Fz or Dsh did lead to a decrease in the stable amounts of Stbm at junctions (*Figure 6B*, *Figure 6—figure supplement 1C–E*, *Figure 6—source data 1*). This could be because Fz and Dsh normally promote Stbm recruitment across cell junctions, which would be consistent with the previously reported stabilisation of Fz across cell junctions by Stbm and Pk (*Warrington et al., 2017*).

## Regulation of Stbm turnover and clustering by Pk

Hyperphosphorylation of Stbm in a *pk* mutant was accompanied by increased Stbm turnover (*Figure 6B*, *Figure 6—figure supplement 1C–E*, *Figure 6—source data 1*), suggesting that Pk may stabilise Stbm by inhibiting its phosphorylation. In contrast, Pk overexpression leads to decreased Stbm turnover (*Figure 6D*, *Figure 6—figure supplement 1F–H*, *Figure 6—source data 1*).

How might Pk regulate Stbm phosphorylation and turnover? One possibility is that the role of Pk is simply to promote complex sorting, which has been shown to occur by Pk destabilising Fz within the same cell, acting via Dsh (*Warrington et al., 2017*). It is possible that when complexes sort out into arrays of the same orientation, clustering of Stbm leads to reduced accessibility to the kinase and thus reduced phosphorylation. We do not favour this 'indirect' model, as hyperphosphorylation

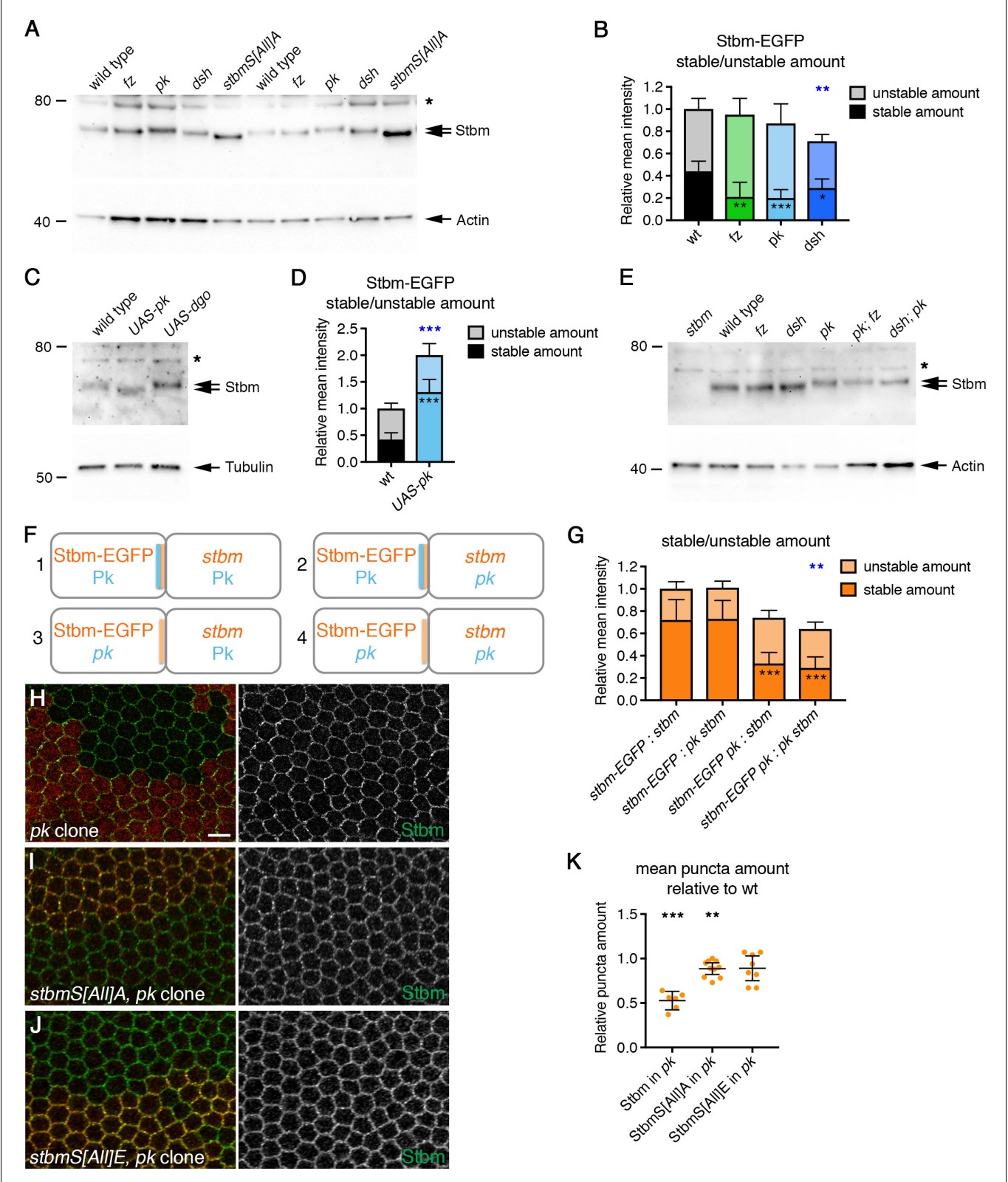

**Figure 6.** Pk reduces Stbm phosphorylation and promotes Stbm stability. (**A,C,E**) Western blots probed with Stbm antibody, of pupal wing extracts. (**A**) Wings from wild-type, $fz^{P21}$, $pk^{pk-sple13}$, $dsh^1$ or $P[acman]-stbmS[All]A$ $stbm^6$ flies at 28 hr APF, two biological replicates are shown for each genotype. (**C**) Wings from wild-type, $Actin-GAL4$, $tub-GAL80^{ts}$, $UAS-pk$ or $Actin-GAL4$, $tub-GAL80^{ts}$, $UAS-dgo$ flies, raised at 29°C for 25 hr APF. (**E**) Wings from wild-type, $fz^{P21}$, $dsh^1$, $pk^{pk-sple13}$, $pk^{pksple13}$; $fz^{P21}$ or $dsh^1$; $pk^{pk-sple13}$ flies at 28 hr APF. The asterisk indicates a non-specific band. Actin (**A,E**) or α-Tubulin

*Figure 6 continued on next page*

*Figure 6 continued*

(**C**) was used as a loading control. (**B,D**) Stable and unstable amounts of Stbm-EGFP in pupal wings after hub-and-spoke FRAP. Flies were (**B**) *P[acman]-stbm-EGFP stbm^6^/+* in a wild-type (n = 11), *fz^P21^* (n = 9), *pk^pk-sple13^* (n = 10) or *dsh^1^* (n = 12) background, at 28 hr APF; (**D**) *P[acman]-stbm-EGFP stbm^6^/+* in a wild-type (n = 9) or *Actin-GAL4, tub-GAL80^ts^, UAS-pk* (n = 8) background, flies raised at 29°C for 25 hr APF. Data are normalised to the total intensity in Stbm-EGFP. Error bars are 95% confidence intervals, and stable amounts (asterisks in dark shaded columns), unstable amounts (asterisks in light shaded columns) or total amounts (blue asterisks above the columns) were compared to the control (Stbm-EGFP) using ANOVA with Dunnett's multiple comparisons test (**B**) or an unpaired t-test (**D**), $p \leq 0.05^*$, $p \leq 0.001^{**}$, $p \leq 0.001^{***}$. (**F,G**) Diagram to illustrate boundary FRAP experiment (**F**), and stable and unstable amounts of Stbm-EGFP on illustrated clone boundaries in 28 hr APF pupal wings (**G**). Flies were *P[acman]-stbm-EGFP stbm^6^* next to *stbm^6^* (column 1, n = 10), *P[acman]-stbm-EGFP stbm^6^* next to *pk^pk-sple13^ stbm^6^* (column 2, n = 10), *P[acman]-stbm-EGFP pk^pk-sple13^ stbm^6^* next to *stbm^6^* (column 3, n = 12) and *P[acman]-stbm-EGFP pk^pk-sple13^ stbm^6^* next to *stbm-EGFP pk^pk-sple13^ stbm^6^* (column 4, n = 10). Error bars are 95% confidence intervals, and ANOVA with Tukey's multiple comparisons test was used to compare stable amounts (asterisks in dark shaded columns), unstable amounts (no significant differences were seen) or total amounts (blue asterisks above the columns) between all genotypes. Comparisons to column one are shown, $p \leq 0.001^{**}$, $p \leq 0.001^{***}$. (**H–J**) 28 hr APF pupal wings from flies carrying *pk^pk-sple13^* clones in a wild-type background (**H**), in a *P[acman]-stbmS[All]A stbm^6^* background (**I**) or in a *P[acman]-stbmS[All]E stbm^6^* background (**J**). Clones immunolabelled for Stbm (green) and marked by loss of β-gal immunolabelling (**H**) or loss of Pk immunolabelling (**I,J**) in red. Scale bar 5 μm. (**K**) Quantitation of mean puncta amount in 28 hr APF pupal wings, shown as a ratio of puncta amount in *pk^pk-sple13^* mutant tissue compared to wild-type tissue in the same wing. Puncta amount is quantitated from wings immunostained for Stbm in a wild-type background (n = 6), a *P[acman]-stbmS[All]A stbm^6^* background (n = 10) or a *P[acman]-stbmS[All]E stbm^6^* background (n = 8). Error bars are 95% confidence intervals. One-sample t-tests were used to determine if the ratio differed from 1.0, $^{**} \leq 0.01$, $^{***}p \leq 0.001$.

DOI: https://doi.org/10.7554/eLife.45107.018

The following source data and figure supplements are available for figure 6:

**Source data 1.** Quantification of Stbm stability and puncta size in core protein mutants.
DOI: https://doi.org/10.7554/eLife.45107.023
**Figure supplement 1.** FRAP analysis of Stbm-EGFP in wings lacking core protein activity.
DOI: https://doi.org/10.7554/eLife.45107.020
**Figure supplement 2.** Pk does not affect phosphorylation and clustering of Stbm phosphorylation site mutants.
DOI: https://doi.org/10.7554/eLife.45107.019
**Figure supplement 3.** Pk cell autonomously regulates Stbm clustering.
DOI: https://doi.org/10.7554/eLife.45107.021
**Figure supplement 4.** Regulation of Stbm clustering by Dsh and Pk.
DOI: https://doi.org/10.7554/eLife.45107.022

of Stbm is not seen in all situations where complexes are thought to be unsorted, for example in *fz* or *dsh* mutants (*Figure 6A*).

An alternative model is that Pk directly regulates Stbm phosphorylation, perhaps by regulating its clustering (see Discussion). To investigate this, FRAP experiments were carried out on clone boundaries, in which Pk activity was present only in the same cell as Stbm, or only in neighbouring cells (*Figure 6F*). In an otherwise wild-type background, Stbm-EGFP strongly accumulates on boundaries next to *stbm* mutant cells, where it is highly stable (*Figure 6G*, column 1, *Figure 6—figure supplement 3A and E–G*, *Figure 6—source data 1*). Stbm-EGFP still accumulates on such boundaries in a *pk* mutant background, but its stability is significantly decreased (*Figure 6G*, column 4, *Figure 6—figure supplement 3D–G*, *Figure 6—source data 1*). If Pk is present only in Stbm-EGFP expressing cells, the phenotype resembles that of the 'wild-type' situation, and Stbm is highly stable (*Figure 6G*, column 2, *Figure 6—figure supplement 3B and E–G*, *Figure 6—source data 1*). In contrast, if Pk is absent only in Stbm-EGFP expressing cells, Stbm-EGFP is unstable (*Figure 6G*, column 3, *Figure 6—figure supplement 3C and E–G*, *Figure 6—source data 1*). This indicates that Pk acts in the same cell to stabilise Stbm and supports a direct role for Pk in regulating Stbm phosphorylation and turnover.

If a major role of Pk was to regulate Stbm phosphorylation, and this phosphorylation regulates Stbm clustering into puncta, we would also expect Stbm phosphomutant clustering to be largely insensitive to loss of Pk. In wild-type wings, loss of Pk causes a reduction in puncta size (*Figure 6H and K*, *Figure 6—source data 1*, *Strutt et al., 2011*). As expected, loss of Pk had less effect on either Stbm phosphomutant or Stbm phosphomimetic puncta (*Figure 6I–K*, *Figure 6—source data 1*). This again supports the conclusion that Pk directly regulates Stbm clustering by modulating the Dco-dependent phosphorylation of Stbm. In contrast, loss of Dsh caused a reduction in phosphomutant and phosphomimetic puncta size, as is also seen in the presence of wild-type Stbm (*Figure 6—*

*figure supplement 4A and B*, *Strutt et al., 2011*). This is consistent with a model in which Fz and Dsh regulate Stbm clustering indirectly by promoting intercellular complex formation.

## Regulation of Dsh junctional localisation by Dco kinase in vivo is independent of Stbm

Having established that Dco regulates Stbm phosphorylation, and this controls the turnover and clustering of Stbm, we next questioned whether these mechanisms are sufficient to explain all the effects of Dco on the core proteins. In particular, does Dco also directly regulate Dsh in any way, or does Dco only regulate Stbm, which then leads to secondary effects on Dsh phosphorylation?

A number of lines of evidence suggest a direct effect of Dco on Dsh. Firstly, we analysed the effect of expressing dominant-negative Dco on EGFP-Dsh turnover. We saw a reduction in overall junctional levels of EGFP-Dsh, leading to a proportionate decrease in the stable amount of Dsh (*Figure 7A*, *Figure 7—figure supplement 1A–C*, *Figure 7—source data 1*). Junctional levels of EGFP-Dsh were also decreased in the absence of Stbm (compare *Figure 7A and B*, *Figure 7—figure supplement 1E–G*, *Figure 7—source data 1*), but were further decreased when dominant-negative Dco was expressed (*Figure 7B*, *Figure 7—figure supplement 1E–G*, *Figure 7—source data 1*). This argues for a role for Dco in regulating Dsh levels at cell junctions independently of Stbm. Furthermore, neither Dsh levels nor Dsh phosphorylation were altered in Stbm phosphomutants (*Figure 7C*, *Figure 3—figure supplement 1E,F,I and J*, *Figure 3—source data 1*), whereas a decrease in Dsh phosphorylation was seen in *dco* hypomorphs (*Figure 7—figure supplement 2A and B*, *Figure 7—source data 1*, *Strutt et al., 2006*).

We then examined whether regulation of Dsh by Dco is important in establishing asymmetry. As discussed above, expression of dominant negative Dco caused a decrease in total levels of Dsh at junctions (*Figure 7A*, *Figure 7—figure supplement 1C*, *Figure 7—source data 1*). However, a decrease in Dsh levels alone is not sufficient to affect asymmetry: halving Dsh dosage halves Dsh levels at junctions (*Figure 7—figure supplement 1C*, *Figure 7—source data 1*), with little effect on the stable fraction (*Figure 7—figure supplement 1D*, *Figure 7—source data 1*) or asymmetry (*Strutt et al., 2016*).

However, two lines of evidence suggest that Dco-mediated phosphorylation of Dsh is functionally important for asymmetry. Firstly, overexpression of Dco caused strong trichome swirling in the adult wing (*Figure 7D*, *Cong et al., 2004*; *Klein et al., 2006*; *Strutt et al., 2006*), and this was accompanied by excess Dsh phosphorylation (*Figure 7—figure supplement 2C and D*, *Figure 7—source data 1*). Notably, these trichome polarity defects were suppressed by halving *dsh* gene dosage, but not *stbm* or *fz* gene dosage (*Figure 7E–H*, *Figure 7—source data 1*). This genetic interaction supports a direct role for Dco in regulating Dsh phosphorylation and core protein asymmetry.

Secondly, we analysed *dsh* mutant flies carrying a *dsh* genomic rescue construct in which eight putative Dco phosphorylation sites were mutated to alanine (*dshST8-GFP*, *Figure 7—figure supplement 3A*, *Strutt et al., 2006*). These flies exhibit only mild defects in trichome polarity (*Strutt et al., 2006*), but core protein asymmetry was not previously examined. As expected, core protein asymmetry was normal in *dsh* mutant flies carrying a wild-type Dsh rescue construct (*Figure 7I and K*, *Figure 7—source data 1*). However, core protein asymmetry was much reduced in *dshST8-GFP* flies (*Figure 7J and K*, *Figure 7—source data 1*), despite levels at junctions being similar to levels of wild-type Dsh (*Figure 7—figure supplement 3B*, *Figure 7—source data 1*). Interestingly, DshST8-GFP has a small but significantly increased rate of turnover at cell junctions, compared to wild-type Dsh-GFP (*Figure 7—figure supplement 3C*, *Figure 7—source data 1*).

Taken together, these results support a model in which Dco-mediated phosphorylation of Dsh regulates its recruitment into junctional complexes and that this is essential for core protein asymmetry.

## Discussion

In this paper, we describe a dual role for CKIε/Dco kinase in regulating planar polarity in the fly pupal wing. In the first case, Dco promotes phosphorylation of Stbm. Stbm phosphorylation acts as a switch, changing Stbm from a stable immobile form that can enter junctional complexes, to an unstable mobile form that can redistribute within cells (*Figure 4C*). Inhibiting Stbm phosphorylation causes an increase in Stbm stability at junctions that prevents sorting of complexes: thus complexes

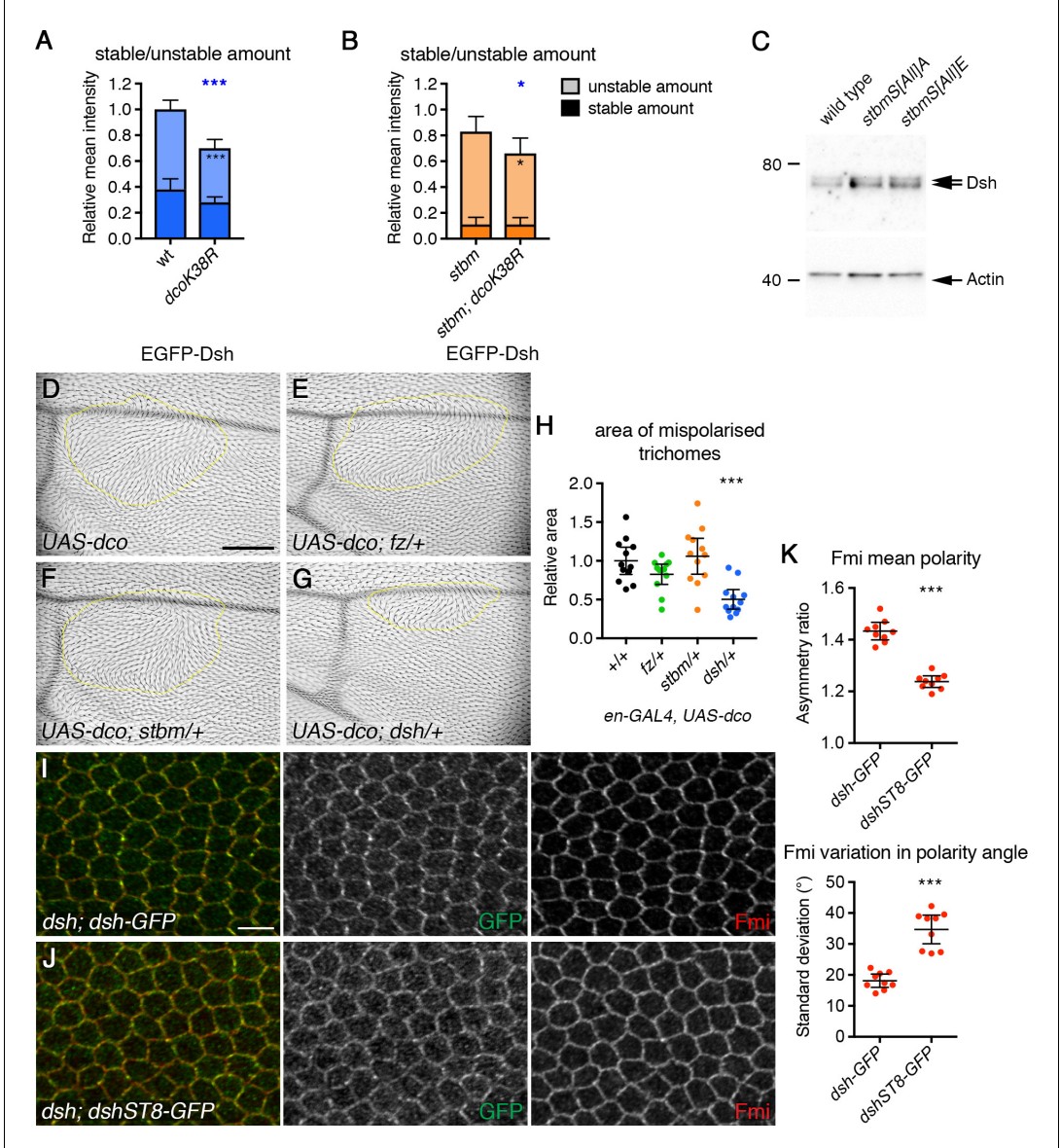

**Figure 7.** Phosphorylation of Dsh by Dco promotes core protein asymmetry. (**A,B**) Stable and unstable amounts of EGFP-Dsh in 28 hr APF pupal wings after hub-and-spoke FRAP. (**A**) Flies were $dsh^{V26}/+$; $P[acman]$-EGFP-dsh/+ (n = 14), and $dsh^{V26}/+$; $P[acman]$-EGFP-dsh/en-GAL4; UAS-dco$^{K38R}/+$ (n = 10). (**B**) Flies were $dsh^{V26}/+$; $P[acman]$-EGFP-dsh stbm$^6$/stbm$^6$ (n = 12), and $dsh^{V26}/+$; $P[acman]$-EGFP-dsh stbm$^6$/stbm$^6$ en-GAL4; UAS-dco$^{K38R}/+$ (n = 7). Wings were imaged in the posterior compartment. Data are normalised to the total intensity in EGFP-Dsh. Error bars are 95% confidence intervals, and unpaired t-tests were used to compare stable amounts (asterisks in dark shaded columns), unstable amounts (asterisks in light shaded columns) or total amounts (blue asterisks above the columns) between samples with and without Dco$^{K38R}$, $p \leq 0.05*$, $p \leq 0.001***$. (**C**) Western blot probed with Dsh antibody, of extracts from 28 hr APF pupal wings from wild-type, $P[acman]$-stbmS[All]A stbm$^6$ or $P[acman]$-stbmS[All]E stbm$^6$ flies. Actin was used as a loading control. (**D–G**) Adult wings expressing en-GAL4, UAS-dco in a wild-type background (**D**) or in flies heterozygous for $fz^{P21}$ (**E**), stbm$^6$ (**F**) or $dsh^{V26}$ (**G**). Regions of the wing with abnormal trichome polarity are outlined in yellow. Scale bar 100 µm. (**H**) Quantitation of trichome swirling in UAS-dco/+; en-GAL4/+ (n = 12), UAS-dco/+; en-GAL4/+; $fz^{P21}/+$ (n = 12), UAS-dco/+; en-GAL4/stbm$^6$ (n = 12) and UAS-dco/$dsh^{V26}$; en-GAL4/+ (n = 12). Graph shows the wing area next to the posterior cross vein in which trichome polarity was abnormal. Data are normalised to the area of the swirl in the UAS-dco control. Error bars are 95% confidence intervals, and ANOVA with Dunnett's multiple comparisons test was used to compare to the UAS-dco control, $p \leq 0.001***$. (**I,J**) 28 hr APF pupal wings from $dsh^{V26}$; dsh-GFP/+ (**I**) or $dsh^{V26}$; dshST8-GFP/+ (**J**). Wings immunolabelled for GFP (green) or Fmi (red). Scale bar 5 µm. (**K**) Quantitation of mean polarity and variation in polarity angle, of 28 hr APF pupal wings immunolabelled for Fmi in $dsh^{V26}$; dsh-GFP/+ (n = 9) or $dsh^{V26}$; dshST8-GFP/+ (n = 9) flies. Error bars are 95% confidence intervals, and samples were compared using an unpaired t-test, ***$p \leq 0.001$.

DOI: https://doi.org/10.7554/eLife.45107.024

The following source data and figure supplements are available for figure 7:

*Figure 7 continued on next page*

*Figure 7 continued*

**Source data 1.** Quantification of Dsh stability and asymmetry.
DOI: https://doi.org/10.7554/eLife.45107.028
**Figure supplement 1.** FRAP analysis of Stbm in *dco* mutant backgrounds.
DOI: https://doi.org/10.7554/eLife.45107.025
**Figure supplement 2.** Regulation of Dsh phosphorylation by Dco.
DOI: https://doi.org/10.7554/eLife.45107.026
**Figure supplement 3.** FRAP analysis of Stbm in *dco* mutant backgrounds.
DOI: https://doi.org/10.7554/eLife.45107.027

are 'locked' in an unsorted state. In contrast, hyperphosphorylation of Stbm destabilises Stbm, allowing it to leave junctions, hence permitting complex sorting. A second role for Dco is to mediate Dsh phosphorylation, which increases Dsh localisation at junctions. Significantly, the effects of Dco on Dsh are independent of Stbm and vice versa.

In our 'cloud model' (*Figure 1C*, *Strutt et al., 2016*), we envisage that multiple binding interactions drive a phase transition from a loosely packed, disordered association of core proteins in non-puncta, towards a highly cross-linked array of complexes within puncta that are all aligned in the same orientation. Stbm is well-placed to be a key component driving such a clustering mechanism, as not only can it multimerise with itself (*Jenny et al., 2003*), but it also has a high stoichiometry within junctions (*Figure 1C*, *Strutt et al., 2016*). Also consistent with a role for Stbm in complex clustering is the observation that Stbm phosphorylation site mutants act as dominant negatives, recruiting wild-type Stbm into non-polarised puncta. Phosphorylation may inhibit a clustering mechanism, due to an increase in negative charge (reviewed in *Wu, 2013*; *Bergeron-Sandoval et al., 2016*; *Boeynaems et al., 2018*).

Interestingly, excess clustering of unphosphorylated Stbm in unsorted complexes is also expected to lead to destabilising feedback interactions with the other core components. When Stbm is unphosphorylated, the increase in Stbm stability is sufficient for Stbm to 'win' over Fmi and Fz. Thus, there is an overall increase in Stbm stability in phosphomutant Stbm puncta, that is accompanied by decreased stability of Fmi and Fz (*Figure 4E–G*).

Pk both promotes Stbm stability and reduces its phosphorylation. A role for Pk in increasing Stbm stability is not surprising, as overexpression of Pk is known to cause excess clustering of the core proteins (*Bastock et al., 2003*; *Tree et al., 2002*). We can envisage a number of mechanisms by which Pk could affect Stbm phosphorylation. *Warrington et al. (2017)* provided evidence that Pk has two roles: firstly, it acts via Dsh to destabilise Fz in the same cell (*Figure 8A*); secondly, it acts via Stbm to stabilise Fz in adjacent cells (*Figure 8B*). In the first case, Pk would promote sorting of complexes, and one possibility is that Stbm is inaccessible to the kinase in sorted complexes, and thus Pk is indirectly reducing Stbm phosphorylation by promoting sorting. Arguing against this, loss of *fz* or *dsh* also abolishes core protein asymmetry, but no hyperphosphorylation is seen. Our boundary FRAP experiments instead support Pk acting directly in the same cell to stabilise Stbm. We therefore propose a mechanism whereby direct binding of Pk to Stbm protects Stbm from phosphorylation.

Interestingly, Stbm has a significantly higher stoichiometry within junctions than Pk (*Strutt et al., 2016*). One possibility is that Stbm forms multimers, and that association of Pk with these multimers causes a conformational change that reduces accessibility to kinase-binding sites. Alternatively, Pk might recruit a phosphatase (albeit no candidates for such a phosphatase are known). The reduced negative charge might then allow Stbm to form higher order structures, which promotes clustering of the entire core protein complex into puncta (*Figure 6—figure supplement 4C*).

Puncta formation in both wild-type and phosphomutants is also dependent on Dsh. Dsh is another a good candidate for promoting clustering as it too can multimerise (*Schwarz-Romond et al., 2007*; *Gammons et al., 2016*), and thus puncta formation may be dependent on clustering on both sides of the complex. Moreover, direct interactions between Stbm and Dsh (*Park and Moon, 2002*; *Bastock et al., 2003*) may promote clustering of unsorted complexes in the absence of phosphorylation.

Feedback models for core protein asymmetry suggest that particular components of the core pathway signal to other components to either stabilise or destabilise them (*Amonlirdviman et al.,*

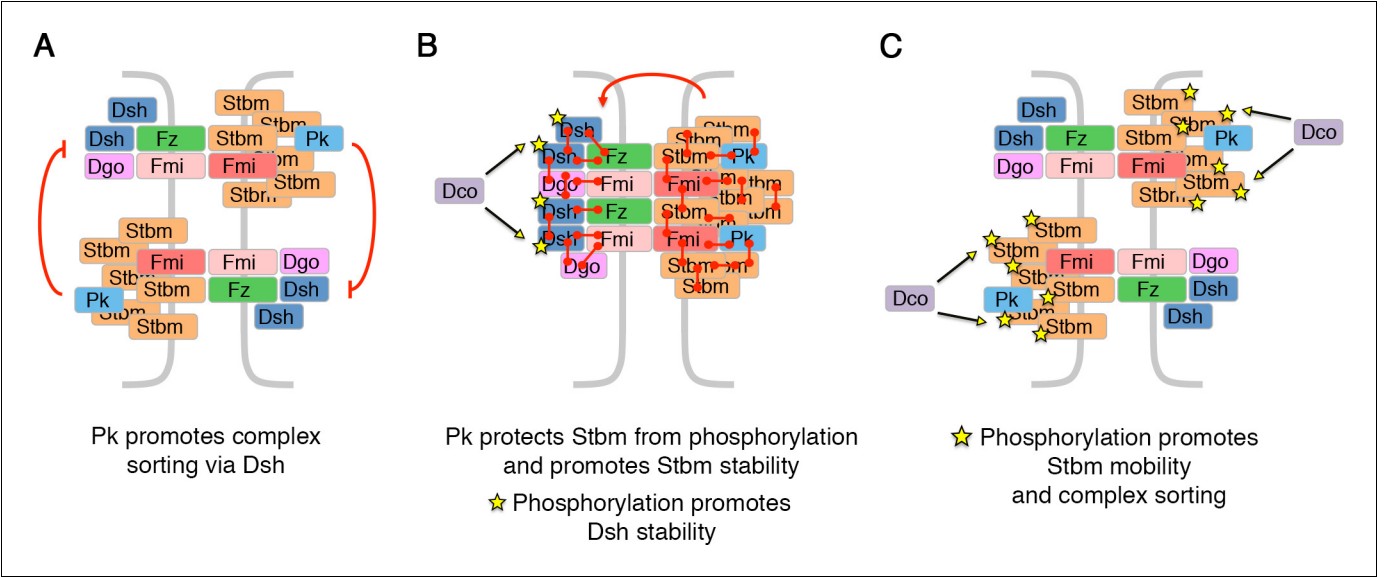

**Figure 8.** Model for how Pk and phosphorylation of Stbm regulate complex sorting and clustering. (**A**) Pk regulates complex sorting by destabilising Fz, in a Dsh-dependent manner (***Warrington et al., 2017***). (**B**) Pk also acts via Stbm to stabilise Fz (red arrow, ***Warrington et al., 2017***). Our new data suggest that Pk promotes Stbm stability by protecting Stbm from phosphorylation by Dco. Furthermore, phosphorylation of Dsh (yellow stars) by Dco promotes Dsh stability. (**C**) When Stbm is hyperphosphorylated (yellow stars) it is more mobile and promotes complex sorting.
DOI: https://doi.org/10.7554/eLife.45107.029

*2005*; ***Burak and Shraiman, 2009***; ***Le Garrec et al., 2006***; ***Schamberg et al., 2010***). An attractive model would be that Fz or Dsh recruits a kinase which phosphorylates Stbm and destabilises complexes of the opposite orientation (***Kelly et al., 2016***; ***Yang et al., 2017***). Consistent with this, a proportion of Dco localises to apicolateral junctions in pupal wings (***Strutt et al., 2006***). However, we do not see any change in Stbm phosphorylation in *fz* or *dsh* mutants, nor are Fz and Dsh required for the hyperphosphorylation of Stbm seen in *pk* mutants. Therefore, we conclude that Stbm phosphorylation is more likely to be constitutive. Such constitutive phosphorylation would be sufficient to keep Stbm mobile and allow complex sorting (***Figure 8C***); and Pk would then counterbalance this and promote complex stability (***Figure 8B***). The balance between Stbm phosphorylation/complex mobility and Pk binding (leading to reduced Stbm phosphorylation) would resolve over time towards a more stable state as complexes segregate to opposite cell edges.

We note that in normal development, Stbm downregulates Pk levels (***Strutt et al., 2013b***). This suggests Pk levels are finely tuned, in order to prevent unrestrained clustering (as seen when Pk is overexpressed).

We also provide evidence that Dco regulates Dsh phosphorylation and junctional levels independently of Stbm. Our findings are consistent with previous observations that Dsh phosphorylation correlates with its recruitment by Fz into junctional complexes (***Axelrod, 2001***; ***Shimada et al., 2001***). The mechanism by which Dsh phosphorylation acts in planar polarity remains to be elucidated, but our data show that *dco* overexpression phenotypes are suppressed by reduced *dsh* gene dosage, and that Dsh phosphomutants have reduced core protein asymmetry in pupal wings. Furthermore, a small but significant decrease in Dsh stability at junctions is observed in Dsh phosphomutants. Overall, our data are consistent with a model in which phosphorylation of Dsh promotes its stable association at junctions (***Figure 8B***).

In summary, we propose that Dco regulates the asymmetric localisation of the core proteins by reciprocal actions on Stbm and Dsh. Dco regulates Stbm phosphorylation and turnover and causes it to leave junctions, while phosphorylation of Dsh by Dco promotes its junctional association.

# Materials and methods

**Key resources table**

| Reagent type (species) or resource | Designation | Source or reference | Identifiers | Additional information |
|---|---|---|---|---|
| Genetic reagent (*Drosophila melanogaster*) | *stbm[6]* | *Wolff and Rubin, 1998*, PMID 9463361 | BDSC:6918; FLYB:FBal0062423; RRID:BDSC_6918 | FlyBase symbol: *Vang[stbm-6]* |
| Genetic reagent (*Drosophila melanogaster*) | *pk[pk-sple13]* | *Gubb et al., 1999*, PMID:10485852 | BDSC:41790; FLYB:FBal0060943; RRID:BDSC_41790 | FlyBase symbol: *pk[pk-sple-13]* |
| Genetic reagent (*Drosophila melanogaster*) | *dsh[V26]* | *Perrimon and Mahowald, 1987*, PMID:3803719 | BDSC:6331; FLYB:FBal0003140; RRID:BDSC_6331 | FlyBase symbol: *dsh[3]* |
| Genetic reagent (*Drosophila melanogaster*) | *dsh[1]* | Bloomington Drosophila Stock Center | BDSC:5298; FLYB FBal0003138; RRID:BDSC_5298 | |
| Genetic reagent (*Drosophila melanogaster*) | *dgo[380]* | *Feiguin et al., 2001*, PMID:11703927 | BDSC:41786; FLYB:FBal0141190; RRID:BDSC_41786 | |
| Genetic reagent (*Drosophila melanogaster*) | *fmi-EGFP* | *Strutt et al., 2016*, PMID:27926869 | | Knock-in of EGFP to the C-terminus of *fmi* in the endogenous locus |
| Genetic reagent (*Drosophila melanogaster*) | *fz-EGFP* | *Strutt et al., 2016*, PMID:27926869 | | Knock-in of EGFP to the C-terminus of *fz* in the endogenous locus |
| Genetic reagent (*Drosophila melanogaster*) | *P[acman]-EGFP-dsh* attP40 (2L) 25C6 | *Strutt et al., 2016*, PMID:27926869 | | *P[acman]* transgene with EGFP recombineered at the N-terminus of *dsh* |
| Genetic reagent (*Drosophila melanogaster*) | *P[acman]-EGFP-dgo* attP40 (2L) 25C6 | *Strutt et al., 2016*, PMID:27926869 | | *P[acman]* transgene with EGFP recombineered at the N-terminus of *dgo* |
| Genetic reagent (*Drosophila melanogaster*) | *P[acman]-stbm* attP40 (2L) 25C6 | *Strutt et al., 2016*, PMID:27926869 | | *P[acman]* transgene |
| Genetic reagent (*Drosophila melanogaster*) | *P[acman]-stbm-EGFP* attP40 (2L) 25C6 | *Strutt et al., 2016*, PMID:27926869 | | *P[acman]* transgene with EGFP recombineered at the C-terminus of *stbm* |
| Genetic reagent (*Drosophila melanogaster*) | *P[acman]-stbm-EGFP* VK1 (2R) 59D3 | This paper | | *P[acman]* transgene with EGFP recombineered at the C-terminus of *stbm* |
| Genetic reagent (*Drosophila melanogaster*) | *P[acman]-EGFP-stbm* attP40 (2L) 25C6 | This paper | | *P[acman]* transgene with EGFP recombineered at the N-terminus of *stbm* |
| Genetic reagent (*Drosophila melanogaster*) | *P[acman]-mApple-stbm* attP40 (2L) 25C6 | This paper | | *P[acman]* transgene with mApple recombineered at the N-terminus of *stbm* |
| Genetic reagent (*Drosophila melanogaster*) | *P[acman]-mApple-dgo* attP40 (2L) 25C6 | This paper | | *P[acman]* transgene with mApple recombineered at the N-terminus of *dgo* |
| Genetic reagent (*Drosophila melanogaster*) | *P[acman]-EGFP-stbmS[5,120,122]A* attP40 (2L) 25C6 | This paper | | *P[acman]* transgene with EGFP at the N-terminus of *stbm*, and with phosphorylation site mutations S[5,120,122]A |
| Genetic reagent (*Drosophila melanogaster*) | *P[acman]-EGFP-stbmS[5,120,122]E* attP40 (2L) 25C6 | This paper | | P[acman] transgene with EGFP at the N-terminus of stbm, and with phosphorylation site mutations S[5,120,122]E |

*Continued on next page*

*Continued*

| Reagent type (species) or resource | Designation | Source or reference | Identifiers | Additional information |
|---|---|---|---|---|
| Genetic reagent (*Drosophila melanogaster*) | *P[acman]-EGFP-stbmS[All]A attP40 (2L) 25C6* | This paper | | P[acman] transgene with EGFP at the N-terminus of stbm, and with phosphorylation site mutations S[5-17, 113-122]A |
| Genetic reagent (*Drosophila melanogaster*) | *P[acman]-stbmS[All]A attP40 (2L) 25C6* | This paper | | P[acman] transgene with phosphorylation site mutations S[5-17, 113-122]A |
| Genetic reagent (*Drosophila melanogaster*) | *P[acman]-stbmS[All]A VK31 (3L) 62E1* | This paper | | P[acman] transgene with phosphorylation site mutations S[5-17, 113-122]A |
| Genetic reagent (*Drosophila melanogaster*) | *P[acman]-stbmS[All]E attP40 (2L) 25C6* | This paper | | P[acman] transgene with phosphorylation site mutations S[5-17, 113-122]E |
| Genetic reagent (*Drosophila melanogaster*) | *P[acman]-stbmS[5-17]A attP40 (2L) 25C6* | This paper | | P[acman] transgene with phosphorylation site mutations S[5-17]A |
| Genetic reagent (*Drosophila melanogaster*) | *P[acman]-stbmS[113-122]A attP40 (2L) 25C6* | This paper | | P[acman] transgene with phosphorylation site mutations S[113-122]A |
| Genetic reagent (*Drosophila melanogaster*) | *P[acman]-stbmS[5-17]E attP40 (2L) 25C6* | This paper | | P[acman] transgene with phosphorylation site mutations S[5-17]E |
| Genetic reagent (*Drosophila melanogaster*) | *P[acman]-stbm S[113-122]E attP40 (2L) 25C6* | This paper | | P[acman] transgene with phosphorylation site mutations S[113-122]E |
| Genetic reagent (*Drosophila melanogaster*) | *P[acman]-stbmS [All]A-EGFP attP40 (2L) 25C6* | This paper | | P[acman] transgene with EGFP at the C-terminus of stbm, and with phosphorylation site mutations S[5-17, 113-122]A |
| Genetic reagent (*Drosophila melanogaster*) | *P[acman]-stbmS [All]E-EGFP VK31 (3L) 62E1* | This paper | | P[acman] transgene with EGFP at the C-terminus of stbm, and with phosphorylation site mutations S[5-17, 113-122]E |
| Genetic reagent (*Drosophila melanogaster*) | *attB-UAS-pk VK20 (3R) 99F8* | This paper | | *pk* gene under control of the *UAS* promoter |
| Genetic reagent (*Drosophila melanogaster*) | *attB-UAS-dgo VK20 (3R) 99F8* | This paper | | *dgo* gene under control of the *UAS* promoter |
| Genetic reagent (*Drosophila melanogaster*) | *CaSpeR-dsh-GFP* | **Axelrod, 2001**, PMID:11358862 | | *dsh* genomic rescue construct, with GFP at the C-terminus |
| Genetic reagent (*Drosophila melanogaster*) | *CaSpeR-dshST8-GFP* | **Strutt et al., 2006**, PMID:16824921 | | *dsh* genomic rescue construct with GFP at the C-terminus, and with phosphorylation site mutations S[236-247]A |
| Genetic reagent (*Drosophila melanogaster*) | *UAS-dco^{K38R}* | **Strutt et al., 2006**, PMID:16824921 | | Dominant negative *dco*, under control of the *UAS* promoter |

*Continued on next page*

*Continued*

| Reagent type (species) or resource | Designation | Source or reference | Identifiers | Additional information |
|---|---|---|---|---|
| Genetic reagent (*Drosophila melanogaster*) | *UAS-dco* | *Sekine et al., 2008*, PMID:18258753 | | *dco* gene under control of the *UAS* promoter |
| Genetic reagent (*Drosophila melanogaster*) | *ptc-GAL4* | Bloomington Drosophila Stock Center | BDSC:2017; FLYB:FBti0002124; RRID:BDSC_2017 | FlyBase symbol: *P{GawB}ptc559.1* |
| Genetic reagent (*Drosophila melanogaster*) | *en-GAL4* | Bloomington Drosophila Stock Center | BDSC:30564; FLYB:FBti0003572; RRID:BDSC_30564 | FlyBase symbol: *P{en2.4-GAL4}e16E* |
| Genetic reagent (*Drosophila melanogaster*) | *Actin-GAL4[25]* | Bloomington Drosophila Stock Center | BDSC:3953; FLYB:FBti0012293; RRID:BDSC_3953 | FlyBase symbol: *P{AyGAL4}25* |
| Genetic reagent (*Drosophila melanogaster*) | *tub-GAL80[ts20]* | Bloomington Drosophila Stock Center | BDSC:7019; FLYB:FBti0027796; RRID:BDSC_7019 | FlyBase symbol: *P{tubP-GAL80ts}[20]* |
| Genetic reagent (*Drosophila melanogaster*) | *Ubx-FLP* | Bloomington Drosophila Stock Center | BDSC:42718; FLYB:FBti0150334: RRID:BDSC_42718 | FlyBase symbol: *P{Ubx-FLP}1* |
| Genetic reagent (*Drosophila melanogaster*) | *hs-FLP[G5] attP2* | Bloomington Drosophila Stock Center | BDSC:55816; FLYB:FBti0160507: RRID:BDSC_55816 | FlyBase symbol: *P{hs-FLP[G5]}attP2* |
| Antibody | rabbit polyclonal anti-Stbm | *Warrington et al., 2013*, PMID:23364328 | RRID:AB_2570077 | 1/1000 (immunolabelling) |
| Antibody | rat polyclonal anti-Stbm | *Strutt and Strutt, 2008*, PMID:18804371 | RRID:AB_2569716 | 1/1000 (immunolabelling) |
| Antibody | affinity purified rabbit polyclonal anti-Fz | *Bastock and Strutt, 2007*, PMID:17652348 | RRID:AB_2801421 | 1/300 (immunolabelling) |
| Antibody | affinity purified rat polyclonal anti-Pk | *Strutt et al., 2013a*, PMID:23487316 | RRID:AB_2569720 | 1/25 (immunolabelling) |
| Antibody | rat polyclonal anti-Dsh | *Strutt et al., 2006*, PMID:16824921 | RRID:AB_2801419 | 1/1000 (immunolabelling) |
| Antibody | mouse monoclonal anti-Fmi #74 | *Usui et al., 1999*, PMID:10490098 | RRID:AB_2619583 | 3 µg/ml (immunolabelling) |
| Antibody | affinity purified rabbit polyclonal anti-GFP | Abcam | Abcam:ab6556; RRID:AB_305564 | 1/4000 (immunolabelling) |
| Antibody | mouse monoclonal anti-ß-gal 40-1a | DSHB | RRID:AB_2314509 | 1/200 (immunolabelling) |
| Antibody | rabbit polyclonal anti-ß-gal | MP Biomedicals/ Cappel | MP Biochemicals: 0855976 (Cappel:55976); RRID:AB_2334934 | 1/4000 (immunolabelling) |
| Antibody | rabbit polyclonal anti-Stbm | *Rawls and Wolff, 2003* PMID:12642492 | | 1/20000 (western blotting) |
| Antibody | affinity purified rabbit polyclonal anti-Dsh | *Strutt et al., 2006*, PMID:16824921 | RRID:AB_2801420 | 1/200 (western blotting) |
| Antibody | mouse monoclonal anti-Actin AC40 | Sigma-Aldrich | Sigma:A4700; RRID:AB_476730 | 1/5000 (western blotting) |

*Continued on next page*

*Continued*

| Reagent type (species) or resource | Designation | Source or reference | Identifiers | Additional information |
|---|---|---|---|---|
| Antibody | mouse monoclonal anti-Tubulin DM1A | Sigma-Aldrich | Sigma:T9026; RRID:AB_477593 | 1/10000 (western blotting) |
| Software, algorithm | ProgResC14 version 1.7.3 | Jenoptik | | |
| Software, algorithm | NIS Elements AR version 4.60 | Nikon | | |
| Software, algorithm | Image Lab version 4.1 | BioRad Laboratories | | |
| Software, algorithm | ImageJ version 2.0.0-r65/1.51 s | https://fiji.sc | | |
| Software, algorithm | Packing Analyzer | *Aigouy et al., 2010*, PMID:20813263 | | |
| Software, algorithm | MATLAB_R2014b | Mathworks | | |
| Software, algorithm | Membrane intensity and Polarity measurement scripts (MATLAB) | *Strutt et al., 2016*, PMID:27926869 | | |
| Software, algorithm | Puncta measurement script (MATLAB) | *Strutt et al., 2019*, PMID: 30661800 | | |
| Software, algorithm | GraphPad Prism version 7.0 c | GraphPad Software, Inc. | | |

## *Drosophila* genetics

*Drosophila melanogaster* flies were grown on standard cornmeal/agar/molasses media at 18°C or 25°C, unless otherwise described.

Fly strains are described in FlyBase. $fz^{P21}$, $stbm^6$, $pk^{pk-sple13}$, $dsh^{V26}$ are null alleles, and $dsh^1$ gives a strong planar polarity phenotype, but functions normally in Wingless signalling (*Axelrod et al., 1998*; *Boutros et al., 1998*).

*P[acman]* constructs (BACPAC resources) were recombineered using standard methods. N-terminal fusions of *P[acman]-EGFP-stbm*, *P[acman]-mApple-stbm* and *P[acman]-mApple-dgo* used plasmid *PL452-N-EGFP* (Addgene) as a source vector, or a modified version *PL452-N-mApple* where mApple replaced EGFP. Gene-specific primers were used to amplify EGFP/mApple and the selection cassette, and the resulting fragment was then recombineered into the relevant *P[acman]* construct, in frame with the open reading frame. The kanamycin cassette was then excised, leaving a single LoxP site between the EGFP/mApple tag and the open-reading frame. Phosphomutants were generated using recombineering with positive-negative selection into *P[acman]-stbm* (*Strutt et al., 2016*), *P[acman]-stbm-EGFP* (*Strutt et al., 2016*) or *P[acman]-EGFP-stbm*. These were exact mutations, leaving no foreign sequence. The open-reading frame of *dgo* was cloned into *attB-pUAST* using standard methods.

*P[acman]* constructs were integrated into the genome via ΦC31-mediated recombination into the *attP40* landing site on 2L, the *VK1* site on 2R or the *VK31* site on 3L. *P[acman]-stbm* lines were recombined or double balanced with $stbm^6$ and *P[acman]-dgo* lines were recombined with $dgo^{380}$, or with $stbm^6$ $dgo^{380}$. *attB-UAS-pk* (*Warrington et al., 2017*) and *attB-UAS-dgo* were integrated into the *VK20* landing site on 3L. Transgenics were made by Genetivision.

*fmi-EGFP* and *fz-EGFP* knock-ins and *P[acman]-stbm*, *P[acman]-stbm-EGFP*, *P[acman]-EGFP-dsh* and *P[acman]-EGFP-dgo* (all in *attP40*) are described in *Strutt et al. (2016)*. Other P element transgene insertions were *CaSpeR-dsh-GFP* (*Axelrod, 2001*), *CaSpeR-dshST8-GFP* (*Strutt et al., 2006*), *UAS-dco^{K38R}* (*Strutt et al., 2006*) and *UAS-dco* (*Sekine et al., 2008*).

Flies were raised at 25°C and dissected or imaged 28 hr after puparium formation (APF), unless otherwise indicated. Flies raised at 29°C were imaged after 25 hr. To avoid dosage compensation effects, females of $dsh^{V26}$/+; *P[acman]-EGFP-dsh*/+ were used. Mitotic clones were induced using

the FLP/FRT system and either *Ubx-FLP* or *hs-FLP*. For expression of *UAS-dco*[K38R] and *UAS-dco*, flies were crossed to *en-GAL4* at 25°C, or *Actin-GAL4, tub-GAL80*[ts] at 29°C. Expression of *attB-UAS-pk* and *attB-UAS-dgo* used *ptc-GAL4* at 25°C or *Actin-GAL4, tub-GAL80*[ts] at 29°C.

Full genotypes for each figure are shown in *Table 1*.

## Adult wing preparations

Adult wings were dehydrated in isopropanol and mounted in GMM (50% methyl salicylate, 50% Canada Balsam), and incubated overnight on a 60°C hot plate to clear. Wings were photographed at 20x magnification. To quantify trichome swirling, ImageJ was used to draw around a region near the posterior cross vein in which trichomes deviated significantly away from their normal orientation. Data were compared using ANOVA with Dunnett's multiple comparisons test.

## Immunolabelling

Pupal wings were dissected at 28 hr after puparium formation (APF) at 25°C. Briefly, pupae were removed from their pupal case and fixed for 25–60 min in 4% paraformaldehyde in PBS, depending on antibody combinations. Wings were then dissected and the outer cuticle removed, and were blocked for 1 hr in PBS containing 0.2% Triton X100 (PTX) and 10% normal goat serum. Primary and secondary antibodies were incubated overnight at 4°C in PTX with 10% normal goat serum, and all washes were in PTX. After immunolabelling, wings were post-fixed in 4% paraformaldehyde in PBS for 30 min. Wings were mounted in 25 µl of PBS containing 10% glycerol and 2.5% DABCO, pH7.5. Wings expressing mApple-tagged proteins were mounted in 12.5 µl Vectashield, as this preserved the fluorescence for longer.

Primary antibodies for immunolabelling were rabbit anti-Stbm (*Warrington et al., 2013*), rat anti-Stbm (*Strutt and Strutt, 2008*), affinity purified rabbit anti-Fz (*Bastock and Strutt, 2007*), affinity purified rat anti-Pk (*Strutt et al., 2013a*), rat anti-Dsh (*Strutt et al., 2006*), mouse monoclonal anti-Fmi (DSHB, *Usui et al., 1999*), rabbit anti-GFP (Abcam cat#6556), mouse monoclonal anti-ß-gal 40-1a (DSHB) and rabbit anti-ß-gal (Cappel).

## Western blotting

For pupal wing westerns, 28 hr APF pupal wings were dissected directly into sample buffer. One pupal wing equivalent was used per lane. For phosphatase treatments, 6 hr APF prepupal wing extracts were made in lysis buffer (50 mM Tris-HCl pH7.5, 150 mM NaCl, 0.5% Triton X-100, 1 x protease inhibitors [Roche]). Lysates were treated with 400 units lambda phosphatase (NEB) for 1 hr at 30°C, before addition of sample buffer.

Western blots were probed with rabbit anti-Stbm (*Rawls and Wolff, 2003*), affinity purified rabbit anti-Dsh (*Strutt et al., 2006*), mouse monoclonal anti-Actin AC-40 (DSHB) and mouse monoclonal anti-α-Tubulin DM1-A (Sigma). SuperSignal West Dura Extended Duration Substrate (Thermo Scientific) was used for detection and a BioRad ChemiDoc XRS + was used for imaging. To quantitate total protein levels, intensities from three or four biological replicates were quantified using ImageJ. Data were compared using ANOVA with Tukey's multiple comparisons test.

For comparing levels of phosphorylated and unphosphorylated Dsh, bands on western blots migrated too close together to quantitate absolute band intensities. ImageJ was used to generate a band profile for each lane, and the maximum values of the phosphorylated and unphosphorylated bands were measured. Data is expressed as a ratio of this maximum intensity, and ratios from four biological replicate samples were compared using unpaired t-tests.

## Imaging of fixed samples

Pupal wings were imaged on a Nikon A1R GaAsP confocal microscope using a 60x NA1.4 apochromatic lens. Wings without clones were imaged posterior to vein 4; wings containing clones were imaged wherever they appeared in the wing. 9 Z-slices separated by 150 nm were imaged at a pixel size of 70–80 nm, and the three brightest slices around apicolateral junctions were selected and averaged for each channel in ImageJ.

Membrane masks and polarity nematics were generated in Packing Analyzer (*Aigouy et al., 2010*). MATLAB scripts were used to calculate mean membrane intensity (*Strutt et al., 2016*).

**Table 1.** List of full genotypes used in each figure.

| Figure | |
|---|---|
| Figure 2A | w; stbm⁶ |

| Figure | |
|---|---|
| *Figure 2A* | *w; stbm$^6$* |
| *Figure 2B* | *w; P[acman]-stbm [attP40] FRT40 stbm$^6$* |
| *Figure 2C* | *w; P[acman]-stbmS[All]A [attP40] FRT40 stbm$^6$* |
| *Figure 2D* | *w; P[acman]-stbmS[All]E [attP40] FRT40 stbm$^6$* |
| *Figure 2E, G, I, L, M* | *y w Ubx-FLP; P[acman]-stbm [attP40] arm-lacZ FRT40 stbm$^6$ / P[acman]-stbmS [All]A [attP40] FRT40 stbm$^6$* |
| *Figure 2F, H, I, L, M* | *y w Ubx-FLP; P[acman]-stbm [attP40] arm-lacZ FRT40 stbm$^6$ / P[acman]-stbmS [All]E [attP40] FRT40 stbm$^6$* |
| *Figure 2J, K* | *w* <br> *w; P[acman]-stbmS[All]A [attP40] FRT40 stbm$^6$* <br> *w; P[acman]-stbmS[All]E [attP40] FRT40 stbm$^6$* |
| *Figure 3A, C* | *y w Ubx-FLP; P[acman]-stbm [attP40] arm-lacZ FRT40 stbm$^6$ / P[acman]-stbmS [All]A [attP40] FRT40 stbm$^6$* |
| *Figure 3B, D* | *y w Ubx-FLP; P[acman]-stbm [attP40] arm-lacZ FRT40 stbm$^6$ / P [acman]-stbmS[All]E [attP40] FRT40 stbm$^6$* |
| *Figure 3G* | *w hs-FLP; P[acman]-EGFP-dgo [attP40] FRT40 dgo$^{380}$ / P[acman]-mApple-dgo [attP40] FRT40 dgo$^{380}$* |
| *Figure 3H* | *w; P[acman]-EGFP-dgo [attP40] FRT40 stbm$^6$ dgo$^{380}$ / P[acman]-mApple-dgo [attP40] FRT40 stbm$^6$ dgo$^{380}$; P[acman-stbmS[All]A [VK31] / P[acman]-stbmS[All]A [VK31] hs-FLP$^{G5}$ [attP2]* |
| *Figure 4B* | *w; P[acman]-stbm-EGFP [attP40] FRT40 stbm$^6$ / +* <br> *w; P[acman]-stbmS[All]A [attP40] FRT40 stbm$^6$ / P[acman]-stbmS[All]A-EGFP [attP40] FRT40 stbm$^6$* <br> *w; P[acman]-stbmS[All]E [attP40] FRT40 stbm$^6$ / stbm$^6$; P[acman]-stbmS[All]E-EGFP [VK31] / +* |
| *Figure 4E* | *w; P[acman]-stbm-EGFP [attP40] FRT40 stbm$^6$* <br> *w; P[acman]-stbmS[All]A-EGFP [attP40] FRT40 stbm$^6$* <br> *w; stbm$^6$; P[acman]-stbmS[All]E-EGFP [VK31]* |
| *Figure 4F* | *w; fmi-EGFP/+* <br> *w; P[acman]-stbmS[All]A [attP40] FRT42 stbm$^6$fmi-EGFP/P[acman]-stbmS[All]A [attP40] FRT40 stbm$^6$* <br> *w; P[acman]-stbmS[All]E [attP40] FRT42 stbm$^6$ fmi-EGFP/P[acman]-stbmS[All]E [attP40] FRT40 stbm$^6$* |
| *Figure 4G* | *w; fz-EGFP/+* <br> *w; P[acman]-stbmS[All]A [attP40] FRT40 stbm$^6$/stbm$^6$; P[acman]-stbmS[All]A [VK31] fz-EGFP / +* <br> *w; P[acman]-stbmS[All]E [attP40] FRT40 stbm$^6$; fz-EGFP / +* |
| *Figure 5A* | *w; P[acman]-stbm-EGFP [attP40] FRT40 stbm$^6$ / +* <br> *w; P[acman]-stbm-EGFP [attP40] FRT40 stbm$^6$ / en-GAL4; UAS-dco$^{K38R}$ / +* <br> *w; P[acman]-stbmS[All]A [attP40] FRT40 stbm$^6$ / P[acman]-stbmS[All]A-EGFP [attP40] FRT40 stbm$^6$* <br> *w; P[acman]-stbmS[Al]A-EGFP [attP40] FRT40 stbm$^6$ / P[acman]-stbmS[All]A [attP40] en-GAL4, stbm$^6$; UAS-dco$^{K38R}$ / +* |
| *Figure 5B* | *w; P[acman]-stbm-EGFP [attP40] FRT40 stbm$^6$ / +* <br> *w UAS-dco/w; P[acman]-stbm-EGFP [attP40] FRT40 stbm$^6$/en-GAL4* <br> *w; P[acman]-stbmS[All]E [attP40] FRT40 stbm$^6$ / stbm$^6$; P[acman]-stbmS[All]E-EGFP [VK31] / +* |

*Table 1 continued on next page*

Table 1 continued

| Figure | |
|---|---|
| *Figure 5C* | *w dsh[1]; P[acman]-stbm-EGFP [attP40] FRT40 stbm[6] / +*<br>*w dsh[1]; P[acman]-stbm-EGFP [attP40]*<br>*FRT40 stbm[6]/ en-GAL4; UAS-dco[K38R] / +* |
| *Figure 6A* | *w*<br>*w; fz[P21]*<br>*w; pk[pk-sple13]*<br>*w dsh[1]*<br>*w; P[acman]-stbmS[All]A [attP40] FRT40 stbm[6]* |
| *Figure 6B* | *w; P[acman]-stbm-EGFP [attP40] FRT40 stbm[6] / +*<br>*w; P[acman]-stbm-EGFP [attP40] FRT40 stbm[6] / +; fz[P21]*<br>*w; P[acman]-stbm-EGFP [attP40]*<br>*FRT42 pk[pk-sple13] stbm[6] / FRT42 pk[pk-sple13]*<br>*w dsh[1]; P[acman]-stbm-EGFP [attP40] FRT40 stbm[6] / +* |
| *Figure 6C* | *w*<br>*w; Actin-GAL4, tub-GAL80[ts] / +; UAS-pk [VK20] / +*<br>*w; Actin-GAL4, tub-GAL80[ts] / +; UAS-dgo [VK20] / +* |
| *Figure 6D* | *w; P[acman]-stbm-EGFP [attP40] FRT40 stbm[6] / +*<br>*w; P[acman]-stbm-EGFP [attP40] FRT40 stbm[6] /*<br>*Actin-GAL4, tub-GAL80[ts]; UAS-pk [VK20] / +* |
| *Figure 6E* | *w; stbm[6]*<br>*w*<br>*w; fz[P21]*<br>*w dsh[1]*<br>*w; pk[pk-sple13]*<br>*w; pk[pk-sple13]; fz[P21]*<br>*w dsh[1]; pk[pk-sple13]* |
| *Figure 6G* | *y w Ubx-FLP; FRT42 stbm[6] P[acman]-stbm-EGFP*<br>*[VK1] / FRT42 stbm[6], Ubi-mRFP-nls*<br>*y w Ubx-FLP; FRT42 stbm[6] P[acman]-stbm-EGFP*<br>*[VK1] / FRT42 pk[pk-sple13] stbm[6], Ubi-mRFP-nls*<br>*y w Ubx-FLP; FRT42 pk[pk-sple13] stbm[6]*<br>*P[acman]-stbm-EGFP [VK1] / FRT42 stbm[6], Ubi-mRFP-nls*<br>*y w Ubx-FLP; FRT42 pk[pk-sple13] stbm[6]*<br>*P[acman]-stbm-EGFP [VK1] / FRT42 pk[pk-sple13] stbm[6], Ubi-mRFP-nls* |
| *Figure 6H, K* | *y w Ubx-FLP; FRT42 arm-lacZ / FRT42 pk[pk-sple13]* |
| *Figure 6I, K* | *y w Ubx-FLP; P[acman]-stbmS[All]A [attP40]*<br>*FRT42 stbm[6] / P[acman]-stbmS[All]A*<br>*[attP40] FRT42 pk[pk-sple13] stbm[6]* |
| *Figure 6J, K* | *y w Ubx-FLP; P[acman]-stbmS[All]E [attP40]*<br>*FRT42 stbm[6] / P[acman]-stbmS[All]E*<br>*[attP40] FRT42 pk[pk-sple13] stbm[6]* |
| *Figure 7A* | *y w dsh[V26] FRT18 / w; P[acman]-EGFP-dsh [attP40]/+*<br>*y w dsh[V26] FRT18 / w; P[acman]-EGFP-dsh*<br>*[attP40] / en-GAL4; UAS-dco[K38R] / +* |
| *Figure 7B* | *y w dsh[V26] FRT18 / w; P[acman]-EGFP-dsh [attP40] stbm[6] / stbm[6]*<br>*y w dsh[V26] FRT18 / w; P[acman]-EGFP-dsh*<br>*[attP40] stbm[6] / en-GAL4, stbm[6]; UAS-dco[K38R] / +* |
| *Figure 7C* | *w*<br>*w; P[acman]-stbmS[All]A [attP40] FRT40 stbm[6]*<br>*w; P[acman]-stbmS[All]E [attP40] FRT40 stbm[6]* |
| *Figure 7D, H* | *w UAS-dco/w; en-GAL4/+* |
| *Figure 7E, H* | *w UAS-dco/w; en-GAL4 / +; fz[P21] / +* |
| *Figure 7F, H* | *w UAS-dco / w; en-GAL4 / stbm[6]* |
| *Figure 7G, H* | *w UAS-dco / w dsh[V26]; en-GAL4 / +* |
| *Figure 7I, H* | *y w dsh[V26] FRT18; pCaSpeR-dsh-GFP / +* |
| *Figure 7J, K* | *y w dsh[V26] FRT18; pCaSpeR-dshST8-GFP / +* |
| | |
| *Figure 2—figure supplement 1A* | *w; P[acman]-EGFP-stbm [attP40] FRT40 stbm[6]* |

Table 1 continued on next page

Table 1 continued

| Figure | |
|---|---|
| *Figure 2—figure supplement 1B* | *w; P[acman]-EGFP-stbmS[5,120,122]A [attP40] FRT40 stbm[6]* |
| *Figure 2—figure supplement 1C* | *w; P[acman]-EGFP-stbmS[5,120,122]E [attP40] FRT40 stbm[6]* |
| *Figure 2—figure supplement 1D* | *w; P[acman]-EGFP-stbmS[All]A [attP40] FRT40 stbm[6]* |
| *Figure 2—figure supplement 1E, I* | *y w Ubx-FLP; P[acman]-EGFP-stbm [attP40] arm-lacZ FRT40 stbm[6] / P[acman]-stbm [attP40] FRT40 stbm[6]* |
| *Figure 2—figure supplement 1F, I* | *y w Ubx-FLP; P[acman]-EGFP-stbmS[5,120,122]A [attP40] FRT40 stbm[6] / P[acman]-EGFP-stbm [attP40] arm-lacZ FRT40 stbm[6]* |
| *Figure 2—figure supplement 1G, I* | *y w Ubx-FLP; P[acman]-EGFP-stbmS[5,120,122]E [attP40] FRT40 stbm[6] / P[acman]-EGFP-stbm [attP40] arm-lacZ FRT40 stbm[6]* |
| *Figure 2—figure supplement 1H, I* | *y w Ubx-FLP; P[acman]-EGFP-stbmS[All]A [attP40] FRT40 stbm[6] / P[acman]-EGFP-stbm [attP40] arm-lacZ FRT40 stbm[6]* |
| *Figure 2—figure supplement 2A* | *w* *w; P[acman]-stbmS[All]A [attP40] FRT40 stbm[6]* *w; P[acman]-stbmS[All]E [attP40] FRT40 stbm[6]* |
| *Figure 2—figure supplement 2B* | *w* *w; P[acman]-stbmS[5-17]A [attP40] FRT40 stbm[6]* *w; P[acman]-stbmS[113-122]A [attP40] FRT40 stbm[6]* |
| *Figure 2—figure supplement 2C, G* | *y w Ubx-FLP; P[acman]-stbmS[5-17]A [attP40] FRT40 stbm[6] / P[acman]-stbm [attP40] arm-lacZ FRT40 stbm[6]* |
| *Figure 2—figure supplement 2D, H* | *y w Ubx-FLP; P[acman]-stbmS[5-17]E [attP40] FRT40 stbm[6] / P[acman]-stbm [attP40] arm-lacZ FRT40 stbm[6]* |
| *Figure 2—figure supplement 2E, G* | *y w Ubx-FLP; P[acman]-stbmS[113-122]A [attP40] FRT40 stbm[6] / P[acman]-stbm [attP40] arm-lacZ FRT40 stbm[6]* |
| *Figure 2—figure supplement 2F, H* | *y w Ubx-FLP; P[acman]-stbmS[113-122]E [attP40] FRT40 stbm[6] / P[acman]-stbm [attP40] arm-lacZ FRT40 stbm[6]* |
| *Figure 3–figure supplement 1A, C, E, G, I* | *y w Ubx-FLP; P[acman]-stbm [attP40] arm-lacZ FRT40 stbm[6]] / P[acman]-stbmS[All]A [attP40] FRT40 stbm[6]* |
| *Figure 3–figure supplement 1B, D, F, H, J* | *y w Ubx-FLP; P[acman]-stbm [attP40] arm-lacZ FRT40 stbm[6] / P[acman]-stbmS[All]E [attP40] FRT40 stbm[6]* |
| *Figure 3—figure supplement 1K* | *w; P[acman]-EGFP-dgo [attP40] FRT40 dgo[380]* |
| *Figure 3—figure supplement 1L* | *w; P[acman]-EGFP-dgo [attP40] FRT40 stbm[6] dgo[380]; P[acman-stbmS[All]A [VK31]* |
| *Figure 4—figure supplement 1A, C, D* | *w; P[acman]-stbm-EGFP [attP40] FRT40 stbm[6] / +* |
| *Figure 4—figure supplement 1A, C, E* | *w; P[acman]-stbmS[All]A [attP40] FRT40 stbm[6] / P[acman]-stbmS[All]A-EGFP [attP40] FRT40 stbm[6]* |
| *Figure 4—figure supplement 1A, C, F* | *w; P[acman]-stbmS[All]E [attP40] FRT40 stbm[6] / stbm[6]; P[acman]-stbmS[All]E-EGFP [VK31] / +* |
| *Figure 4—figure supplement 2A-D* | *w* |
| *Figure 4—figure supplement 2B, D* | *w; P[acman]-stbmS[All]A [attP40] FRT40 stbm[6] / +* |
| *Figure 4—figure supplement 2C, D* | *w; P[acman]-stbmS[All]E [attP40] FRT40 stbm[6] / +* |
| *Figure 5—figure supplement 1A* | *w* *w; FRT82 dco[2] / FRT82 dco[5B2.6]* |
| *Figure 5—figure supplement 1B* | *w* *w; Actin-GAL4, tub-GAL80[ts] / +; UAS-dco[K38R] / +* |
| *Figure 5—figure supplement 2A-C* | *w; P[acman]-stbm-EGFP [attP40] FRT40 stbm[6]/+* *w; P[acman]-stbm-EGFP [attP40] FRT40 stbm[6] / en-GAL4; UAS-dco[K38R] / +* *w; P[acman]-stbmS[All]A [attP40] FRT40 stbm[6] / P[acman]-stbmS[All]A-EGFP [attP40] FRT40 stbm[6]* *w; P[acman]-stbmS[Al]A-EGFP [attP40] FRT40 stbm[6] / P[acman]-stbmS[All]A [attP40] en-GAL4, stbm[6]; UAS-dco[K38R] / +* |

*Table 1 continued*

| Figure | |
| --- | --- |
| *Figure 5—figure supplement 2D-F* | *w; P[acman]-stbm-EGFP [attP40] FRT40 stbm$^6$ / +*<br>*w UAS-dco / w; P[acman]-stbm-EGFP [attP40] FRT40 stbm$^6$ / en-GAL4*<br>*w; P[acman]-stbmS[All]E [attP40] FRT40 stbm$^6$ / stbm$^6$; P[acman]-stbmS[All]E-EGFP [VK31] / +* |
| *Figure 5—figure supplement 2G-I* | *w dsh$^1$; P[acman]-stbm-EGFP [attP40] FRT40 stbm$^6$ / +*<br>*w dsh$^1$; P[acman]-stbm-EGFP [attP40] FRT40 stbm$^6$ / en-GAL4; UAS-dco$^{K38R}$ / +* |
| *Figure 6—figure supplement 2A* | *w*<br>*w; pk$^{pk-sple13}$* |
| *Figure 6—figure supplement 2B* | *w; P[acman]-stbmS[All]E [attP40] FRT40 stbm$^6$*<br>*w; P[acman]-stbmS[All]E [attP40] FRT42 pk$^{pk-sple13}$ stbm$^6$*<br>*w; P[acman]-stbmS[All]A [attP40] FRT40 stbm$^6$*<br>*w; P[acman]-stbmS[All]A [attP40] FRT42 pk$^{pk-sple13}$ stbm$^6$* |
| *Figure 6—figure supplement 1A* | *y w Ubx-FLP; P[acman]-EGFP-stbm [attP40] FRT40 stbm$^6$ / P[acman]-mApple-stbm [attP40] FRT40 stbm$^6$* |
| *Figure 6—figure supplement 1B* | *y w Ubx-FLP; P[acman[-EGFP-Stbm [attP40] FRT40 ptc-GAL4 stbm$^6$ / P[acman]-mApple-Stbm [attP40] FRT40 stbm$^6$; UAS-Pk [VK20] / +* |
| *Figure 6—figure supplement 1C-E* | *w; P[acman]-stbm-EGFP [attP40] FRT40 stbm$^6$/+*<br>*w; P[acman]-stbm-EGFP [attP40] FRT40 stbm$^6$/+; fz$^{P21}$*<br>*w; P[acman]-stbm-EGFP [attP40] FRT42 pk$^{pk-sple13}$ stbm$^6$ / FRT42 pk$^{pk-sple13}$*<br>*w dsh$^1$; P[acman]-stbm-EGFP [attP40] FRT40 stbm$^6$ / +* |
| *Figure 6—figure supplement 1F-H* | *w; P[acman]-stbm-EGFP [attP40] FRT40 stbm$^6$ / +*<br>*w; P[acman]-stbm-EGFP [attP40] FRT40 stbm$^6$ / Actin-GAL4, tub-GAL80$^{ts}$; UAS-pk [VK20] / +* |
| *Figure 6—figure supplement 3A, E-G* | *y w Ubx-FLP; FRT42 stbm$^6$ P[acman]-stbm-EGFP [VK1] / FRT42 stbm$^6$, Ubi-mRFP-nls* |
| *Figure 6—figure supplement 3B, E-G* | *y w Ubx-FLP; FRT42 stbm$^6$ P[acman]-stbm-EGFP [VK1] / FRT42 pk$^{pk-sple13}$ stbm$^6$, Ubi-mRFP-nls* |
| *Figure 6—figure supplement 3C, E-G* | *y w Ubx-FLP; FRT42 pk$^{pk-sple13}$ stbm$^6$ P[acman]-stbm-EGFP [VK1] / FRT42 stbm$^6$, Ubi-mRFP-nls* |
| *Figure 6—figure supplement 3D, E-G* | *y w Ubx-FLP; FRT42 pk$^{pk-sple13}$ stbm$^6$ P[acman]-stbm-EGFP [VK1]/ FRT42 pk$^{pk-sple13}$ stbm$^6$, Ubi-mRFP-nls* |
| *Figure 6—figure supplement 4A* | *w dsh$^{V26}$ FRT19A/y w Ubx-FLP FRT19A; P(acman)-StbmS(All)A [attP40] FRT40 stbm[6]* |
| *Figure 6—figure supplement 4B* | *w dsh$^{V26}$ FRT19A / y w Ubx-FLP FRT19A; P(acman)-StbmS(All)E [attP40] FRT40 stbm[6]* |
| *Figure 7—figure supplement 1A-D* | *y w dsh$^{V26}$ FRT18 / w; P[acman]-EGFP-dsh [attP40] / +*<br>*y w dsh$^{V26}$ FRT18 / w; P[acman]-EGFP-dsh [attP40] / en-GAL4; UAS-dco$^{K38R}$ / +*<br>*y w dsh$^{V26}$ FRT18; P[acman]-EGFP-dsh [attP40] / +* |
| *Figure 7—figure supplement 1E-G* | *y w dsh$^{V26}$ FRT18 / w; P[acman]-EGFP-dsh [attP40] stbm$^6$ / stbm$^6$*<br>*y w dsh$^{V26}$ FRT18 / w; P[acman]-EGFP-dsh [attP40] stbm$^6$ / en-GAL4, stbm$^6$; UAS-dco$^{K38R}$ / +* |
| *Figure 7—figure supplement 2A, B* | *w*<br>*w; FRT82 dco$^2$ / FRT82 dco$^{5B2.6}$* |
| *Figure 7—figure supplement 2C, D* | *w*<br>*UAS-dco / w; en-GAL4 / +* |
| *Figure 7—figure supplement 3B, C* | *y w dsh$^{V26}$ FRT18; pCaSpeR-dsh-GFP / +*<br>*y w dsh$^{V26}$ FRT18; pCaSpeR-dshST8-GFP / +* |

DOI: https://doi.org/10.7554/eLife.45107.030

Polarity magnitude (maximum asymmetry ratio on a cell-by-cell basis) and variation in polarity angle were also calculated using MATLAB scripts (*Strutt et al., 2016*).

To compare puncta between wild-type and mutant tissue, a MATLAB script was used to select puncta using the same threshold value in wild-type and mutant regions of the same wings (*Strutt et al., 2019*). Puncta number per unit area was calculated, as well as mean puncta amount (puncta area multiplied by mean puncta intensity).

Values for control and mutant regions of the same wings (for clones) were expressed as a ratio and compared using one sample t-tests; or were compared between images using unpaired t-tests or ANOVA for more than two genotypes. For all experiments n = number of wings.

## Live imaging

For live imaging, a small piece of cuticle was removed from over the pupal wings of 28 hr APF pupae, and the exposed wing was mounted in a drop of Halocarbon 700 oil in a glass-bottomed dish. For FRAP analysis, images were $256 \times 256$ pixels, with a pixel size of 100 nm, and a pinhole of 1.2 AU. 'Hub-and-spoke' ROIs of $3–4 \ \mu m^2$ were selected, that covered a vertex and three half-cell edges. Alternatively elliptical ROIs were selected to bleach puncta (ROIs $1–1.5 \ \mu m^2$) or entire junctions between two cells on a clone boundary (ROIs $2–2.5 \ \mu m^2$). Three pre-bleach images were taken at two frames/sec, and ROIs were then bleached using a 488 nm Argon laser at 80% with eight passes (1 s total time), which resulted in 60–75% bleaching. Immediately following bleaching, five images were taken at 5 s intervals, followed by 10 images at 10 s intervals and 26 images at 15 s intervals. Laser power was adjusted to maintain constant power between different imaging sessions. If only EGFP was being imaged, a long pass GFP filter was used. If mRFP was present in the stock, EGFP was detected using a 525–550 band pass filter.

## FRAP processing

For data analysis, ImageJ was used to manually reselect up to six bleached regions in each image for each time point. The laser off background was subtracted, and the values were corrected for acquisition bleaching and normalised against the average of the prebleach values. Data were then plotted on an xy graph using Prism (v7 Graphpad), bleached regions within the same wing were averaged and a one-phase exponential curve was fitted for each wing. Multiple wings were then combined and an exponential association curve was fitted. An extra-sum-of-squares F test was used to compare curve plateaux (y[max]), and stable fractions were calculated as 1-y[max].

For hub-and-spoke and boundary FRAP experiments, the stable amount of protein was calculated by measuring the intensity of the ROIs from the three pre-bleach images, and averaging per wing. The intensity was then corrected for distance from the coverslip as previously described (*Strutt et al., 2016*), and this value was then multiplied by the stable fraction (1-y[max]) for each wing. The stable amounts were then averaged across wings.

Overall junctional intensities, and stable and unstable amounts were compared between genotypes using unpaired t-tests, or one-way ANOVA for more than two genotypes. Post-hoc tests were used to compare individual samples: Dunnett's multiple comparison test was used to compare the control to the rest of the genotypes in the experiment; Tukey-Kramer's multiple comparison test to compare all genotypes within an experiment; and Holm-Šídák's multiple comparison test was used to compare genotypes pair-wise.

Each experiment was performed on multiple wings from different pupae, which represent biological replicates (n = number of wings). For each wing, 4 ROIs were selected for FRAP analysis, and these were treated as technical replicates and were averaged per wing to produce a y[max] and a stable amount per wing.

Based on the mean intensity and standard deviation of a control set of wings, we calculated that a sample size of 6 wings per genotype would allow detection of differences of 20% in the means, in a pair-wise comparison, with a power of 0.8 and $\alpha$ 0.05 (using G*Power). As standard deviations were larger for some genotypes, we aimed for 10 wings per genotype. Data was excluded if the ROI recovery curve failed the 'replicates test for lack of fit' in GraphPad Prism, or if the wing moved out of focus during the course of imaging.

## Acknowledgements

We thank the Bloomington Drosophila Stock Center for fly stocks, Tanya Wolff and the Developmental Studies Hybridoma Bank for antibodies, and Genetivision for generating transgenics. Natalia Bulgakova, Katie Fisher, Simon Fellgett and Melissa Gammons are thanked for comments on the manuscript, and the fly room staff for excellent technical support. Imaging was performed in the Wolfson Light Microscopy Facility. The work was funded by Wellcome Senior Fellowship awards to DS [100986/Z/13/Z and 210630/Z/18/Z] and an MRC studentship (grant numbers G0900203-1/1 and G1000405-1/1).

## Additional information

### Funding

| Funder | Grant reference number | Author |
|---|---|---|
| Wellcome Trust | 100986/Z/13/Z | Helen Strutt David Strutt |
| Medical Research Council | G0900203-1/1 | Jessica Gamage |
| Wellcome Trust | 210630/Z/18/Z | Helen Strutt David Strutt |
| Medical Research Council | G1000405-1/1 | Jessica Gamage |

The funders had no role in study design, data collection and interpretation, or the decision to submit the work for publication.

### Author contributions

Helen Strutt, Formal analysis, Investigation, Methodology, Writing—original draft, Writing—review and editing; Jessica Gamage, Investigation, Writing—review and editing; David Strutt, Conceptualization, Supervision, Funding acquisition, Methodology, Writing—review and editing

### Author ORCIDs

Helen Strutt  https://orcid.org/0000-0003-4365-2271
David Strutt  https://orcid.org/0000-0001-8185-4515

### Decision letter and Author response

Decision letter https://doi.org/10.7554/eLife.45107.033
Author response https://doi.org/10.7554/eLife.45107.034

## Additional files

### Supplementary files

• Transparent reporting form
DOI: https://doi.org/10.7554/eLife.45107.031

### Data availability

All data generated or analysed during this study are included in the manuscript and supporting files.

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
