## [Decision Letter]

Thank you for submitting your article "Reciprocal action of Casein Kinase Iε on core planar polarity proteins regulates clustering and asymmetric localisation" for consideration by *eLife*. Your article has been reviewed by three peer reviewers, and the evaluation has been overseen by a Reviewing Editor and Utpal Banerjee as the Senior Editor. The reviewers have opted to remain anonymous.

As you will see, all of the reviewers were impressed with the importance and novelty of your work. I am including the three reviews at the end of this letter, as there are a variety of specific and useful suggestions in them. We appreciate that the reviewers' comments cover a broad range of suggestions for improving the manuscript. We look forward to receiving your revised manuscript.

*Reviewer #1:*

The manuscript examines the role the Casein kinase Dco plays in the regulation of core PCP proteins, especially Stbm and Dsh. The study confirms recent findings the Dco affects Stbm phosphorylation, and extends these by examining the effects of phosphorylation site mutants on *Drosophila* wing PCP, and on Stbm polarization, stability at junctions, puncta size, etc. Thus they go well beyond the study of Kelly et al., 2016 especially in vivo, and in detail some of their results disagree. Intriguingly, phosphomutant and phosphomimetic Stbm both have abnormal and even dominant negative effects on PCP and are associated in decreased and increased junctional stability, respectively. The authors do a nice job incorporating this into their model.

Previous work had suggested that Stbm phosphorylation could be regulated by Dsh and Fz. However, the authors find little evidence of this in vivo from fz and dsh mutants. Instead, they find a strong effect from Pk. While they do not have a firm understanding of how this works, they provide some persuasive arguments that it is likely fairly direct, rather than through simple assembly of the PCP complex or through Fz or Dsh.

Finally, they go on to show that Dco also has an independent effect on Dsh levels and phosphorylation.

While this paper lacks the zing of recent studies on polarity-inducing feedback mechanisms, it is certainly a solid study that extends previous work on the role of PCP component phosphorylation. I think it deserves publication.

*Reviewer #2:*

In this study, Strutt et al. investigate the role of phosphorylation in the establishment of planar cell polarity (PCP) in *Drosophila*. Using immunofluorescence, live imaging, FRAP, biochemistry, and genetic mosaic approaches, they provide evidence that Stbm is less stable and more mobile when phosphorylated, and conversely, more stable when unphosphorylated. In contrast to Stbm, Dsh was found to be more stable when phosphorylated. They also provide evidence that Pk negatively regulates Stbm phosphorylation. Their findings suggest that Casein Kinase Ie mediates Stbm phosphorylation, confirming other reports, and extend this finding by providing genetic evidence that Casein Kinase Ie phosphorylates both Stbm and Dsh independently. They propose a model for how modifying differential protein stability could be means to regulate sorting of PCP proteins.

Mechanisms that regulate the establishment and maintenance of PCP is of high interest to the fields of polarity and developmental biology. Although Stbm/Vang and Dsh/Dvl have been known to be phosphorylated for some time, and that their phosphorylation is important for PCP, it was not known how phosphorylation regulates PCP protein dynamics and function. By providing further support that Casein Kinase Ie-mediates phosphorylation of Stbm and Dsh and by demonstrating that phosphorylation alters PCP protein stability and sorting, this study provides important and substantial contributions to the field. A strength of the study is the use of tissue extracts from transgenic and mutant lines to measure phosphorylation states (as opposed to transfected cells), but the reliance on very subtle band shifts as the sole means to detect phosphorylation is a weakness. The manuscript text is generally well written, however the data and figures could have been presented more clearly. While statistics used were appropriate and thorough, the quantification is often confusing and needs to be made more transparent. Though the findings are significant and impactful with some mechanistic detail, there are concerns that the work does not go far enough to extend current knowledge and has limited accessibility in its current state for the broad readership of *eLife*.

Major comments:

- Inclusion of a table of *Drosophila* genotypes used with direct reference of how they are labeled/referred to in figures their phenotypes would be very helpful.

- No biochemical evidence is shown to support that direct binding of Pk to Stbm protects Stbm from phosphorylation (as discussed in the Discussion, fourth paragraph).

- Figure 1 lays the foundation for all subsequent figures and could be presented more clearly for a broader audience. For example, the schematic in A makes sense to a PCP specialist but needs more context for someone less familiar with the field. Zoomed images for E-F will help better visualize trichome orientation, and a reorganization of E-M to better pair panels of the same genotypes would help. Additionally, an overlay of polarity vectors for I-J would be helpful as Stbm asymmetry is subtle and not apparent to the untrained eye.

- The authors state that "only small puncta were observed in Stbm phosphomimetic flies, and non-puncta material also increased (Figure 1J). However, in the Stbm phosphomutant, Stbm still clustered into large puncta, despite the loss of asymmetry (Figure 1I)." Quantification of the puncta sizes in Figure 1I-J and Figure 2A-D are lacking and necessary as many of the future conclusions rely on these observations.

- Figure 2G-H is a critical piece of data to support the authors model that -P promotes sorting of opposite PCP complexes, and should be presented more convincingly. Panel H shows Diego localized to both sides of cell junctions in Stbm phosphomutants, suggesting PCP complexes of opposite orientations are mixed. According to the model, puncta in wild type flies represent sorted PCP complexes, but is this observed during intermediate stages of polarization? Diego is clearly segregated in the fully polarized control in panel G, but at earlier stages, do they see puncta along a single junction that are sorted in opposite orientations? This control would strengthen the argument that it is the sorting of opposite complexes within puncta that is defective in their mutants. According to the model, stbmS(AII)E complexes/puncta should be sorted and thus, that data should be included stbmS(AII)E phosphomimetic for G-H panels.

- Figure 3. Stable to unstable amounts should be displayed separately to better see the changes, it is hard to compare the relative amounts as currently displayed (this holds true for all FRAP data).

- Figure 5. The western blot bands in A demonstrating a shift in Stbm are not obvious. This is main evidence that Pk influences Stbm phosphorylation, but it's not wholly convincing. In general, detecting changes in -P solely by band shifts is a general weakness of the overall approach since these differences are very subtle, hard to quantify, and open to other interpretations. Although this is probably beyond scope of this paper, generating a phospho-specific Ab or performing mass spec analyses of -P sites would go a long way towards strengthening the results and lending support to the overall model.

- The model in Figure 7 would benefit from inclusion of Dco. In addition, do the authors think that Pk destabilizes Fz via inhibiting Dco-dependent -P of Dsh?

*Reviewer #3:*

The manuscript by Strutt and colleagues addresses the mechanisms by which protein complexes become asymmetrically distributed on cell junctions, a process important for establishing planar cell polarity. The authors identified an important role of CK1 in phosphorylating core proteins and thereby regulating their clustering/membrane association. This work gives interesting new insights into the role that phosphorylation plays in establishing the asymmetry of core proteins at cell junctions. The manuscript is well written and the data are convincing for the most part. I have a few concerns, which I expect could be addressed within a reasonable time by the authors:

Figure 1J, L The authors claim that in Stbm phosphomimetic flies only small puncta are observed, whereas in Stbm phosphomutant flies, Stbm clusters in large puncta. The puncta are not obvious from these images. Higher magnification images should be shown and puncta size needs to be quantified. Moreover, the authors should mention whether *stbm*[6] mutants still express Stbm protein, i.e. do these stainings show Stbm[6] and phosphomutant Stbm or only the phosphomutant Stbm?

Figure 2G, H I can see that in the Stbm phosphomutant flies Diego forms "non-polarized" puncta, however, I do not see evidence that these puncta co-localize with Stbm, as the authors claim.

Figure 3A. The authors should briefly explain the concept of the 'hub-and-spoke' FRAP methods, so that readers do not have to look up the reference in order to understand the experiment. Moreover, images of the FRAP experiment should be shown (e.g. in the supplement) to allow readers to evaluate the data.

Figure 3 FRAP analysis allows to estimate the mobile fraction of a protein, and the authors use this method to show that phosphorylation of Stbm increases its mobile fraction. It is however, less clear how the authors infer from the FRAP analysis that phosphorylation suppresses clustering in complexes. While complex formation might be one mechanism to reduce the mobile fraction of a protein, it is not the only one. The authors should elaborate on this issue.

---

## [Author Response]

Reviewer #2:[…] A strength of the study is the use of tissue extracts from transgenic and mutant lines to measure phosphorylation states (as opposed to transfected cells), but the reliance on very subtle band shifts as the sole means to detect phosphorylation is a weakness. The manuscript text is generally well written, however the data and figures could have been presented more clearly. While statistics used were appropriate and thorough, the quantification is often confusing and needs to be made more transparent. Though the findings are significant and impactful with some mechanistic detail, there are concerns that the work does not go far enough to extend current knowledge and has limited accessibility in its current state for the broad readership of eLife.Major comments:- Inclusion of a table of *Drosophila* genotypes used with direct reference of how they are labeled/referred to in figures their phenotypes would be very helpful.

We have added a table of all genotypes associated with each figure.

- No biochemical evidence is shown to support that direct binding of Pk to Stbm protects Stbm from phosphorylation (as discussed in the Discussion, fourth paragraph).

We have discussed other possible models for how Pk might regulate Stbm phosphorylation in the manuscript, but we found evidence arguing against the other models that we considered. We agree however that we don't have direct evidence for the idea that binding of Pk to Stbm protects Stbm from phosphorylation; this is why we put the associated model figure in the supplementary data rather than in the main manuscript. However we think it is useful to suggest hypotheses about the mechanism, as this is then a basis for future studies. In the future we will be looking for direct evidence for this model, but such experiments are beyond the scope of the current manuscript.

- Figure 1 lays the foundation for all subsequent figures and could be presented more clearly for a broader audience. For example, the schematic in A makes sense to a PCP specialist but needs more context for someone less familiar with the field. Zoomed images for E-F will help better visualize trichome orientation, and a reorganization of E-M to better pair panels of the same genotypes would help. Additionally, an overlay of polarity vectors for I-J would be helpful as Stbm asymmetry is subtle and not apparent to the untrained eye.

These are all good points, and we have added extra introductory panels to the first figure. The adult wings in (now in Figure 2 and Figure 2—figure supplement 1) have been replaced with higher magnification images. We have also overlaid polarity vectors on the images in Figure 2 and Figure 2 supplements 1 and 2. Finally, we have also included higher magnification images of puncta in these genotypes, in response to reviewer 3.

- The authors state that "only small puncta were observed in Stbm phosphomimetic flies, and non-puncta material also increased (Figure 1J). However, in the Stbm phosphomutant, Stbm still clustered into large puncta, despite the loss of asymmetry (Figure 1I)." Quantification of the puncta sizes in Figure 1I-J and Figure 2A-D are lacking and necessary as many of the future conclusions rely on these observations.

We have now included measurements of puncta number and size in Figure 2. We previously attempted to threshold images using Stbm immunolabelling, using the same threshold for wild-type and mutant regions of the same wing. However as overall junctional levels of Stbm increase in both the phosphomutant and the phosphomimetic, this was problematic and resulted in apparent increased values in both cases. We have now got round this issue by using Fmi immunolabelling to threshold the images: as Fmi co-localises with Stbm and junctional levels of Fmi do not change in the phosphomutants, this successfully identifies puncta without "spreading" into neighbouring non-puncta regions. The major finding is that puncta *number* is similar in wild-type and phosphomutant; suggesting a similar degree of "punctateness"; while the phosphomimetic is less punctate. The amount of Stbm in puncta (puncta area multiplied by puncta intensity) is however slightly lower in both the phosphomutant and phosphomimetic. We think the smaller puncta size in phosphomutant and phosphomimetic may be due to the fact that negative feedback limits the growth of individual puncta (as seen in our FRAP experiments using Fmi-EGFP and Fz-EGFP).

We have modified the manuscript to take into account these quantitations; in particular we have stated that the phosphomimetic forms fewer puncta rather than that it forms only small puncta.

However, the main conclusion still stands: that the Stbm phosphomimetic clusters into puncta more readily than the phosphomimetic and thus the coupling between puncta formation and asymmetry is lost.

- Figure 2G-H is a critical piece of data to support the authors model that -P promotes sorting of opposite PCP complexes, and should be presented more convincingly. Panel H shows Diego localized to both sides of cell junctions in Stbm phosphomutants, suggesting PCP complexes of opposite orientations are mixed. According to the model, puncta in wild type flies represent sorted PCP complexes, but is this observed during intermediate stages of polarization? Diego is clearly segregated in the fully polarized control in panel G, but at earlier stages, do they see puncta along a single junction that are sorted in opposite orientations? This control would strengthen the argument that it is the sorting of opposite complexes within puncta that is defective in their mutants. According to the model, stbmS(AII)E complexes/puncta should be sorted and thus, that data should be included stbmS(AII)E phosphomimetic for G-H panels.

We have not looked at Dgo localisation at earlier developmental stages, but we speculate that the sorting would be less good than at 28 hr APF. We expect that sorting *within* puncta would be reduced; and that most puncta would contain complexes in both orientations rather than seeing puncta along a single junction that are sorted in opposite orientations.

We also don't predict that individual Stbm phosphomimetic complexes/puncta should be sorted. Our data suggest that Stbm phosphomimetics behave like *pk* mutants, in which Stbm is hyperphosphorylated. In our model hyperphosphorylation promotes Stbm mobility and thus the puncta that do form are unstable. This means that complexes are continually forming and unforming and that complexes of the same orientation fail to accumulate. It would be interesting to test this directly, but we are unable to do the twin clone experiment with our current reagents, as *P[acman]-stbmS[All]E, P[acman]-mApple-dgo* and *P[acman]-EGFP-dgo* are in the same *attP* site.

- Figure 3. Stable to unstable amounts should be displayed separately to better see the changes, it is hard to compare the relative amounts as currently displayed (this holds true for all FRAP data).

We agree that it is not easy to compare the unstable amounts in the current graphs. However, if we plot stable and unstable amounts separately, we will "lose" the other important data regarding total intensity. We have however added extra supplementary figures for all FRAP data, showing the total intensity, stable amount and unstable amounts as separate plots.

- Figure 5. The western blot bands in A demonstrating a shift in Stbm are not obvious. This is main evidence that Pk influences Stbm phosphorylation, but it's not wholly convincing. In general, detecting changes in -P solely by band shifts is a general weakness of the overall approach since these differences are very subtle, hard to quantify, and open to other interpretations. Although this is probably beyond scope of this paper, generating a phospho-specific Ab or performing mass spec analyses of -P sites would go a long way towards strengthening the results and lending support to the overall model.

We agree that the band shifts are subtle, and that is why we have shown duplicates of the critical samples. We have now also included an extra supplementary figure panel (Figure 6—figure supplement 1A) in response to reviewer 1, to show that the decreased migration of Stbm in *pk* mutants is sensitive to phosphatase.

We also agree that phospho-specific antibodies would be nice, but also that this is beyond the scope of this paper. Furthermore, we note that other researchers have made phospho-specific antibodies (e.g. for Dsh, Yanfeng et al., 2011) and it is difficult to get enough sensitivity on westerns in an in vivo context.

- The model in Figure 7 would benefit from inclusion of Dco. In addition, do the authors think that Pk destabilizes Fz via inhibiting Dco-dependent -P of Dsh?

We have added Dco to the model diagram (now in Figure 8).

In our previous work, we proposed two mechanisms by which Pk could destabilise Fz. In the first, Dsh would normally protect Fz from endocytosis (e.g. by promoting multimerisation); and that Pk binding to Dsh could block this protective function. In the second mechanism, we proposed that binding of Pk to Fz/Dsh could lead to recruitment of an enzyme that mediates a change in post-translational modification of Dsh or Fz. In either model, we could envisage either a direct role for Dco (e.g. Pk locally inhibits Dco function or promotes a phosphatase that opposes Dco activity) or indirect roles (e.g. multimerised Dsh is a better target for Dco or is protected from a phosphatase, and Pk opposes multimerisation thus indirectly leading to a change in the Dco-mediated phosphorylation of Dsh). However, we think we might be speculating beyond what the data can really support if we included such ideas in the manuscript.

Reviewer #3:[…] I have a few concerns, which I expect could be addressed within a reasonable time by the authors:Figure 1J, L The authors claim that in Stbm phosphomimetic flies only small puncta are observed, whereas in Stbm phosphomutant flies, Stbm clusters in large puncta. The puncta are not obvious from these images. Higher magnification images should be shown and puncta size needs to be quantified. Moreover, the authors should mention whether stbm[6] mutants still express Stbm protein, i.e. do these stainings show Stbm[6] and phosphomutant Stbm or only the phosphomutant Stbm?

We have now shown higher magnification images of the puncta in Figure 2 and Figure 2—figure supplements 1 and 2. We have also provided quantitations of puncta number and size (see response to reviewer 2).

We have also clarified in the manuscript that the clones of the phosphorylation site mutants are generated in the absence of endogenous Stbm.

Figure 2G, H, I can see that in the Stbm phosphomutant flies Diego forms "non-polarized" puncta, however, I do not see evidence that these puncta co-localize with Stbm, as the authors claim.

We have now shown in Figure 3—figure supplement 1 that EGFP-Dgo co-localises in puncta with the other core proteins, in both a wild-type and a Stbm phosphomutant background.

Figure 3A. The authors should briefly explain the concept of the 'hub-and-spoke' FRAP methods, so that readers do not have to look up the reference in order to understand the experiment. Moreover, images of the FRAP experiment should be shown (e.g. in the supplement) to allow readers to evaluate the data.

We have added more explanation to the text to better explain the methodology. We have also added some example FRAP images (pre-bleach, immediately post-bleach and after 500 sec recovery) for the first experiment, and the entire FRAP datasets are in the source data.

Figure 3 FRAP analysis allows to estimate the mobile fraction of a protein, and the authors use this method to show that phosphorylation of Stbm increases its mobile fraction. It is however, less clear how the authors infer from the FRAP analysis that phosphorylation suppresses clustering in complexes. While complex formation might be one mechanism to reduce the mobile fraction of a protein, it is not the only one. The authors should elaborate on this issue.

We agree that in general a reduction in the mobile fraction of a protein can be caused by mechanisms other than complex formation; for instance other explanations could be reduced vesicular trafficking or tethering to another structure such as the cytoskeleton. However, our model is based not only on the FRAP data but on the fact that we see changes in cluster formation (puncta) in our images (now quantitated in the revised manuscript). Taken together, we think our current working model is the most plausible. We would prefer not to add further speculation to the Discussion, which is already quite long.